# Gemstones: A Model Suite for Multi-Faceted Scaling Laws

**Sean McLeish**[1]*, **John Kirchenbauer**[1], **David Yu Miller**[1], **Siddharth Singh**[1]
**Abhinav Bhatele**[1], **Micah Goldblum**[2], **Ashwinee Panda**[1], **Tom Goldstein**[1]
[1] University of Maryland, [2] Columbia University

## Abstract

Scaling laws are typically fit using a family of models with a narrow range of frozen hyperparameter choices. In this work we study scaling laws using multiple architectural shapes and hyperparameter choices, highlighting their impact on resulting prescriptions. As a primary artifact of our research, we release the *Gemstones*: an open-source scaling law dataset, consisting of over 4000 checkpoints from transformers with up to 2 billion parameters and diverse architectural shapes; including ablations over learning rate and cooldown. Our checkpoints enable more complex studies of scaling, such as analyzing the relationship between width and depth. By examining our model suite, we find that the prescriptions of scaling laws can be highly sensitive to the experimental design process and the specific model checkpoints used during fitting.
**Code**: github.com/mcleish7/gemstone-scaling-laws

## 1   Introduction

Existing works on scaling laws often restrict Transformer architectures to a small range of width-depth ratios [Porian et al., 2024], train on a small number of tokens, and fix training hyperparameters such as cooldown schedule across training runs [Hoffmann et al., 2022]. These design choices, in turn, can dramatically influence the resulting scaling laws. If a scaling law is sensitive to such design choices, then it may only be useful for practitioners implementing similar setups to those that produced the scaling law. In practice, practitioners often take guidance from scaling laws that assume completely different design choices than their own implementation, often without understanding to degree to which these choices may impact optimal scaling.

In this work, we produce a vast array of model checkpoints for studying how model design and model selection impact scaling laws. Our models, called the *Gemstones* because they are loosely based on scaled-down variants of the Gemma architecture, vary in their parameter count, width/depth ratio, training tokens, learning rates, and cooldown schedules. By fitting scaling laws to these checkpoints, we confirm that scaling law parameters and interpretations indeed depend strongly on the selection of models and fitting procedure used, and we quantify the degree to which these decisions impact predictions. By exploiting the variation among our

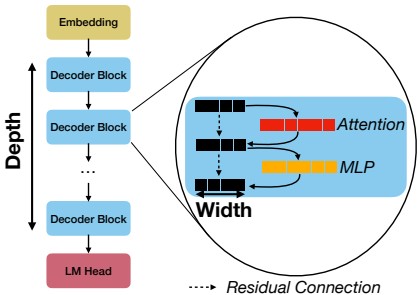

Figure 1: **The meaning of width and depth.** We visualize a standard transformer architecture, highlighting the "width" as the size of the hidden dimension and the "depth" as the number of transformer blocks.

---

*Correspondence to: `smcleish@umd.edu`.

39th Conference on Neural Information Processing Systems (NeurIPS 2025).

model checkpoints, we also analyze the impact of architectural shape across loss, benchmark performance and training time with findings consistent with design choices we see in industry models. Our contributions are summarized as follows:

**We open-source more than 4000 checkpoints cumulatively trained on over 10 trillion tokens.** The models we provide are diverse across architectural and training hyperparameter axes, enabling more granular studies than previous work (see Figure 2).

**We highlight the fragility and common pitfalls of prior scaling laws.** There are many decisions to make when choosing points to fit scaling laws that significantly change the slope of the law (see Table 1).

**We analyze the impact of model shape on loss, benchmarks and training time.** We find that although deep models achieve lower loss and benchmark error when measured using floating point operations, they require significantly more training time when using typical open-source parallelism frameworks (see Figures 4, 7 and 8).

## 2   Related Work

Scaling laws address the trade-off between parameter count and number of training tokens, attempting to find the minimum loss possible for a language model with a constrained floating point operation (FLOP) budget. The body of work on scaling laws is vast. Therefore, while we provide a brief overview of key prior work here to contextualize our contributions, we also include an extended literature review in Appendix B.

**Design Choices for Scaling Laws**   Scaling laws often treat model design and training as if it has a single dimension (parameter count), while, in practice, training is sensitive to many choices. Notably, Hoffmann et al. [2022] find significantly different fitted laws (Equation (1)) compared to Kaplan et al. [2020]. Pearce and Song [2024] and Porian et al. [2024] attribute most of this discrepancy to the choice to exclude embedding parameters from the parameter count, both showing one law can be transformed into the other via controlled changes. Kaplan et al. [2020] justify excluding embedding parameters by showing that non-embedding parameters have a cleaner relationship with test loss. Scaling laws are also commonly included in many large model releases [Hu et al., 2024, Bi et al., 2024, Dubey et al., 2024].

Choshen et al. [2024] collect both loss and benchmark performance metrics for a multitude of models and offer a practitioner's guide to fitting scaling laws. Notably, they suggest that 5 models are ample to fit a scaling law, and that you should omit the early part of training when fitting, because those early steps don't follow the same scaling behavior and can skew the results. In contrast, Li et al. [2024c] demonstrate that varying the tokens-per-parameter ratio and relying on limited grid searches when fitting scaling laws can lead to large variations in results. Hägele et al. [2024] suggest that a constant learning rate plus cooldown is preferable to a cosine learning rate schedule as all intermediate checkpoints can be used for fitting. The authors also find that stochastic weight averaging should be encouraged in scaling law analysis as it tends to lead to better models. Furthermore, Inbar and Sernau [2024] observe that FLOPs cannot be used to predict wall-clock time nor memory movement, and suggest that fast-training architectures may be preferred over those prescribed by scaling laws.

There are multiple works analyzing whether scaling laws can be used to predict downstream performance. Bhagia et al. [2024] first predict task-specific loss and then use this to predict performance on the task, using a sigmoidal function to map from loss to accuracy. Gadre et al. [2024] predict top-1 error, fitting a power law function on the perplexity of the model to predict error. Gadre et al. [2024] also look at the impact of overtraining, finding scaling laws that extrapolate with the amount of overtraining. Dey et al. [2023] analyze the trade off of inference FLOPs and training FLOPs using scaling laws to prescribe training configurations that balance training and inference. Unfortunately, both Dey et al. [2023] and Biderman et al. [2023] train on the Pile [Gao et al., 2020] which has since been taken down due to copyright, leaving a gap for a model collection in the open literature.

**The Role of Model Shape**   Another line of research specifically studies the interplay between model width and model depth; for clarity we visualize our working definitions for these dimensions in Figure 1. Levine et al. [2020] find that, for large models, optimal depth grows logarithmically with width. Henighan et al. [2020] find there is an optimal aspect ratio for each modality they study which

gives maximum performance: for example, they find 5 to be optimal for math models. Team et al. [2024b] compare two 9 billion parameter models and find the deeper model outperforms the wider one consistently across benchmarks. Unfortunately, the authors are vague about the specific details of this result. Petty et al. [2024] claim small (<400M) transformers have diminishing benefits from depth. Brown et al. [2022] show that in some cases shallower models can beat their parameter-equivalent deep models on tasks for encoder-decoder transformer architectures. These results differ from Kaplan et al. [2020] who suggest aspect ratio is not a determining factor for final loss. Tay et al. [2022] show that downstream performance strongly depends on shape when finetuning but pretraining perplexity does not. Alabdulmohsin et al. [2024] study the impact of width and depth for encoder-decoder vision transformers, using their laws to create a smaller transformer model which has competitive downstream performance when compared with much larger models. The architecture found in this study has since been used by Beyer et al. [2024] in PaliGemma. Concurrently, Zuo et al. [2025] study the impact of width and depth in hybrid architectures, finding that a deeper 1.5B model can match or even outperform 3B and 7B models.

As discussed above, the literature on how the aspect ratio of a LLM affects its performance and scaling characteristics is simultaneously extensive but somewhat inconclusive. While we do not presume to fully answer every question in this space, the experiments we describe in the rest of this work make progress on how to understand the results of prior studies and the impacts of certain architecture choices in a fully open, reproducible, and extensible way.

## 3 Designing Our Scaling Laws

We discuss the design of our scaling laws, including model selection, the choice of learning rate, and curve fitting schemes in the subsequent sections and in greater detail in Appendix A.

**Architecture.** To reduce the search space of all possible models, we add some constraints, each of which are either based on precedent from a popular model series like Gemma [Team et al., 2024a,b], Llama [Touvron et al., 2023b], Pythia [Biderman et al., 2023], or practical considerations such as hardware details (see Appendix A).

Within these constraints, we search the set of feasible models within target parameter count groups $50M, 100M, 500M, 1B$ and $2B$ with a tolerance of $\pm5\%$. At smaller scales we train up to 5 models at diverse widths and depths. At large parameter counts we train only 3 models, aiming for one "standard" aspect ratio (similar to existing models), one "wide" model, and one "deep" model. We visualize the models we choose to train in Figure 2 overlaid with a selection of existing models from prior work. In the Appendix, we plot the entire discrete set of all possible models under our constraints (Figure 10). Our 22 different models range from $50M$ to $2B$ parameters, spanning 11 widths from 256 to 3072 and 18 depths from 3 to 80.

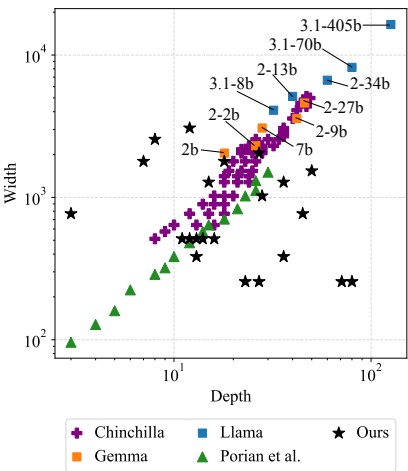

Figure 2: **Distribution of prior scaling law models, industry models, and our models in terms of width and depth.** Prior work (purple and green) and industry models (blue and orange) mostly lie on a fixed width-depth line.

**Polishing the Gemstones.** For the main set of training runs, we train each model for $350B$ tokens of Dolma 1.7 [Soldaini et al., 2024] data. We target a total batch size of 4 million tokens following [Touvron et al., 2023b, Dubey et al., 2024, Bai et al., 2023], with a context length of 2048 and a world batch size of 2048 sequences. Following Hägele et al. [2024] and Hu et al. [2024], we use a linear learning rate warm up over 80 million tokens, and then train at a constant learning rate, which we adjust for model size as described in Appendix A.1.

In service of future research based on our model suite, we open source checkpoints for all models at 2 billion token intervals, amounting to over $4,000$ checkpoints in total. We also open source the fitting code and logged metrics for all runs.

We perform ablations over both cooldown and learning rate. For the cooldown ablation, we take the checkpoints saved every 10 billion tokens for the the first 100 billion tokens of training and cool these down, creating a second set of models which have had their learning rate annealed to 0 linearly. Specifically, we cool each model down by training for a further $10\%$ of the total tokens which it has seen during training, i.e. our cooled down set of models have training budgets ranging from 11 to 110 billion tokens. We also ablate our choice of learning rate by running all models for 100 billion tokens with half of the learning rate we use for our main analysis. The full details of the scalable learning rate and parameter initialization scheme–designed to enable hyperparameter transfer across model sizes and aspect ratios–are provided in Appendix A.1.

**Training Details** We train with AdamW [Loshchilov and Hutter, 2017] with $\beta$ parameters 0.9 and 0.95 and a weight decay of 0.1. We do not apply weight decay to the bias or normalization parameters. All models are trained with tensor parallelism [Singh and Bhatele, 2022, Singh et al., 2024] over multiple nodes of AMD MI250X GPUs. To the best of our knowledge, this makes the Gemstone suite of models the largest collection trained on AMD GPUs.

## 3.1 Fitting Scaling Laws

We fit scaling laws using methods similar to approach 1 and 3 from Chinchilla [Hoffmann et al., 2022]. We fit all laws using the log perplexity of all trained models on a sample of 100 million tokens from a fixed, held-out validation set from the training distribution. We also collect log perplexity values for a range of open source models [Team et al., 2024a,b, Touvron et al., 2023b, Dubey et al., 2024, Yang et al., 2024a,b] on the same validation data to allow for a comparison between our predictions and a selection of widely used models. We design a specialized FLOP counting function as we find that simple rules of thumb (e.g., FLOPs per token= $6 \times parameters$ [Hoffmann et al., 2022]) do not accurately account for differences in FLOPs between extremely wide and narrow architectures. We discuss this further and present our function in Appendix M.

Following Porian et al. [2024], we plot the Epoch AI Replication [Besiroglu et al., 2024] of Chinchilla [Hoffmann et al., 2022] on all plots and use the coefficients for Kaplan plotted by Porian et al. [2024] which were extracted from the original paper [Kaplan et al., 2020].

**A More Robust Approach to Fitting Compute-Optimal Laws.** The first approach in Hoffmann et al. [2022] fits a scaling law by plotting the loss against FLOPs for a family of models with a range of parameter counts (but relatively consistent aspect ratio, see Figure 2) while varying dataset size, then fitting a line to the pareto-optimal architecture for each FLOP count (see Figure 3). Following Hoffmann et al. [2022], we refer to this as "Approach 1". As we use a constant learning rate, we can use all recorded validation losses to fit our law. Hoffmann et al. [2022] and Kaplan et al. [2020] select model shapes so densely that they have a near-optimal architecture at each FLOP count. This works when all architectures lie in a 1D space (parameterized by parameter count), as each model is optimal in some FLOP regime, and the lower envelope is densely populated. However, in our two dimensional exploration (varying width and depth), some models are never optimal, and the ones that are do not densely populate the envelope. We therefore develop a novel fitting method to accommodate sampling strategies like ours that result in regions of lower data density.

**Our New Method: The Convex Hull.** We fit a lower *convex hull* to our loss curves. This hull is only supported by a sparse set of optimal models. Fitting on only the vertices of this hull naturally excludes sub-optimal models that lie above the convex hull of optimality, and as we show in Section 4, this makes the resulting scaling law far more robust to model selection choices. We provide a mathematical definition of our approach in Appendix D.

**Why We Skip Approach 2.** Another method to fit scaling laws is to put model runs into isoFLOP bins and choose the best parameter count in each bin. Hoffmann et al. [2022] call this "Approach 2". Our 2-dimensional set of models do not finely cluster into isoFLOP bins, meaning our data is not easily amenable to Approach 2, hence we exclude this approach from our analysis. Hu et al. [2024] and Li et al. [2024c] also eschew this approach.

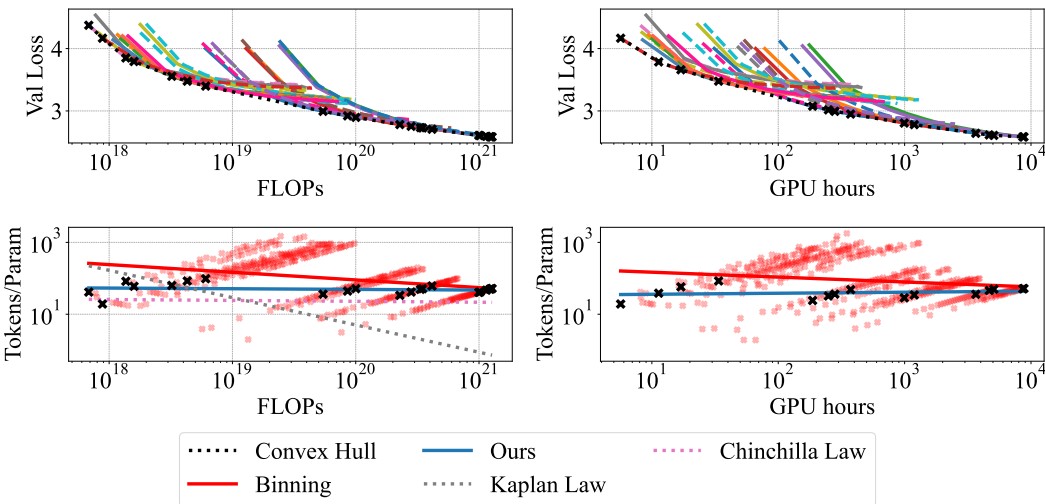

Figure 3: **Approach 1 prescriptions.** Row one: Validation loss over FLOPs (left) and GPU hours (right) for the first 100 billion tokens of training. We use Approach 1 to find the optimal points on the convex hull in each setting, marked with **black crosses**. Row two: We fit a line to the tokens per parameter of empirically optimal models and find a slightly higher, but still constant, tokens per parameter prescription than Hoffmann et al. [2022]. Hoffmann et al. [2022]'s Approach 1 creates 250 logarithmically-spaced FLOPs bins per order of magnitude, and in **red** we plot the minimizers over these bins, and the scaling law fitted to these minimizers (binning). Clearly, their Approach 1 is not well-suited for our data, and our convex hull approach is better when we select fewer models to fit our law on. Extended plot in Figure 20.

**Prescribing Optimal Tokens Per Parameter by Fitting Power Laws.** The final approach described by Hoffmann et al. [2022] is to fit a parametric formula to the loss values with the ansatz

$$L(p, T) = \frac{A}{p^\alpha} + \frac{B}{T^\beta} + \varepsilon \tag{1}$$

where $p$ is parameter count and $T$ is tokens. We fit our models using L-BGFS [Liu and Nocedal, 1989] with a Huber loss ($\delta = 10^{-4}$) between the empirical log loss and the model prediction, and use multiple initializations following Besiroglu et al. [2024]. We ablate to check that our fitting procedure is robust to the size of the grid of initializations and the choice of delta in Appendix L.4.

## 4 Experiments

In Section 4.1, we use our new convex hull fitting method to make a scaling law for the compute-optimal tokens-to-parameters ratio, and compare this to the prescription from our fitted power laws. We show that many design choices such as the learning rate schedule can significantly impact these prescribed scaling laws in Section 4.2. In Section 4.3, we analyze the Gemstones loss values over multiple datasets and connect our analysis to benchmarks. Finally, we perform an analysis over time taken to train instead of over FLOPs in Section 4.4.

### 4.1 Sizing Up Our Scaling Laws Against Prior Laws and Industry Models

**Approach 1.** In Figure 3 (row one), we see our validation losses plotted as both a function of FLOPs (left) and GPU hours (right) for the first 100 billion tokens of training. We calculate GPU hours from the average recorded optimizer step time for each model.

**Our convex hull fits the data better than prior approaches.** Hoffmann et al. [2022]'s Approach 1 creates 250 logarithmically-spaced FLOPs bins per order of magnitude and then uses the models that achieve the best loss in each FLOPs bin to fit the scaling law (a line). However, for our data, their approach does not work very well because it includes many points that are strictly suboptimal with respect to the minimal loss envelope. Our convex hull method omits these points, and fits the line

Table 1: **We demonstrate the variability in fitting scaling laws by resampling our data many different ways.** The slope can be viewed as the exponent in the power law relationship $parameters = constant \cdot compute^{exponent}$. Grouping by fitting approach and choice to include embeddings, in the final column 'Delta' we show the change in slope produced by the ablations against the corresponding base law fit on the full set of hot data. Values with an absolute magnitude greater than 0.05 are highlighted in orange, and those exceeding 0.1 are highlighted in red. We see that the reduced sampling has a large impact on the slope of the law and that Approach 1 is more sensitive than Approach 3. We plot these prescriptions in Figure 14 and show this table with embeddings excluded from the parameter count in Table 2.

| Tokens | Cooldown | LR Ablation | Embeddings | Slope | Delta |
|---|---|---|---|---|---|
| Hoffmann et al. [2022] | | | | 0.5126 | |
| **Approach 1 (w/ Embeds)** | | | | | |
| all | ✗ | ✗ | ✓ | 0.4579 | |
| ≤ 100b | ✗ | ✗ | ✓ | 0.4994 | 0.0415 |
| > 120b | ✗ | ✗ | ✓ | 0.7987 | 0.3408 |
| all | ✗ | ✓ | ✓ | 0.5131 | 0.0552 |
| all | ✓ | ✗ | ✓ | 0.5970 | 0.1391 |
| **Approach 3 (w/ Embeds)** | | | | | |
| all | ✗ | ✗ | ✓ | 0.6965 | |
| ≤ 100b | ✗ | ✗ | ✓ | 0.6986 | 0.0021 |
| > 120b | ✗ | ✗ | ✓ | 0.7515 | 0.0550 |
| all | ✗ | ✓ | ✓ | 0.6740 | -0.0225 |
| all | ✓ | ✗ | ✓ | 0.6992 | 0.0027 |
| **Approach 3 – Chinchilla Reduced Sampling** | | | | | |
| all | ✗ | ✓ | ✓ | 0.6328 | -0.0636 |
| all | ✗ | ✗ | ✓ | 0.6315 | -0.0649 |
| Hoffmann et al. [2022] | ✗ | ✗ | ✓ | 0.6123 | 0.0997 |

with far fewer points. The asymptotic flatness of power law curves makes trying to fit a scaling law an ill-conditioned optimization problem. Our novel convex hull approach is specifically crafted to reduce this variance and our results suggest that when optimal points are sparse, our approach can be used to obtain a more reliable fit (red vs black crosses in Figure 3)

In Figure 3 (row two), we present the prescriptions from our scaling laws for tokens per parameter. We see that the tokens per parameter prescription of our Approach 1 fitting is close to constant, as in Hoffmann et al. [2022], but suggests a slightly larger optimal tokens per parameter ratio than their law. We extend this plot showing predicted total parameters, tokens, and over multiple ablations in Appendix L and we give a more detailed plot of each model's individual validation loss in Appendix K. In Appendix F, we show a leave-one-out analysis over models when fitting both Approach 1 and 3.

## 4.2 Fragility and Pitfalls of Scaling Laws

To demonstrate the sensitivity of scaling laws to design choices, we fit laws with various assumptions and model selection rules. To provide compute-optimal parameter count prescriptions, we use equation 4 from Hoffmann et al. [2022], which we restate in Equation (6) for the convenience of the reader.

In Table 1 we show the optimal predictions of multiple possible laws fitted on different subsets of our data. The "slope" column can be viewed as the exponent in the power law relationship between compute and parameters. In the final column "Delta", we show the change in slope produced by the ablations against the corresponding base law fit on the full set of hot data, grouping by fitting approach and choice to include embeddings. We also plot these prescriptions with a FLOPs x-axis in Figure 14.

One particular dimension of variability we wish to highlight briefly here is the interplay between model selection and the derived law. To do this, we select 5 models from Gemstones that have an analogous model in Hoffmann et al. [2022] (using data extracted by Besiroglu et al. [2024]) with similar parameter count and aspect ratio and then we select Gemstones checkpoints with token counts nearly matching the Hoffmann points. We call this "Chinchilla Reduced Sampling" and

fit scaling laws to both of these sub-sampled datasets. We find that fitting Hoffmann's data using this reduced sampling results in an increased slope relative to fitting on all data. Meanwhile this subsampling reduces the slope of the line fit on Gemstones.[2] This highlights that scaling law fitting can be quite sensitive to seemingly innocuous changes in model selection for both the variable aspect ratio Gemstones models as well as the simpler model family selected by Hoffman.

Between Table 1 and Table 2 we present the complete results from our series of ablations. Table 1 shows the results of fitting laws while including embedding parameter count, which both Pearce and Song [2024] and Porian et al. [2024] find to be a primary explanation of the discrepancies between the prescriptions found by Kaplan et al. [2020] and Hoffmann et al. [2022]. Then in Table 2 we report results when not including the embedding parameter count. We also show the impact of fitting on our cooldown and learning rate ablation datasets in turn, seeing that both choices have a noticeable impact on the prescription for optimal parameter count. Finally, we remove checkpoints from our data to simulate having only trained for 100 billion tokens or only having data for token counts greater than 120 billion, seeing a greater impact than when fitting on our ablation data.

### 4.3 Modeling Performance on Different Validation Sets and Downstream Benchmarks

In Figure 4 we plot the loss of the $\geq 500M$ Gemstone models on Dolma, FineWeb, FineWeb-Edu and DCLM-baseline data [Soldaini et al., 2024, Penedo et al., 2024, Li et al., 2024b]. We see that when varying the data distribution on which we compute validation loss, although the loss changes for the Gemstones, it is equivalent to a y-axis shift from Dolma; the relative ordering of models remains unchanged. Of particular note is the fact that the deeper models consistently provide a lower loss.

Next, following Penedo et al. [2024], we benchmark our Gemstone models on MMLU [Hendrycks et al., 2020], WinoGrande [Sakaguchi et al., 2021], OpenBook QA [Mihaylov et al., 2018], ARC [Clark et al., 2018], CommonSense QA [Talmor et al., 2018], PIQA [Bisk et al., 2020], SIQA [Sap et al., 2019] and HellaSwag [Zellers et al., 2019]. Specifically, we benchmark the models at 10 billion token intervals during training. We show the benchmark accuracy at selected token counts in Figure 8.

**Predicting Benchmark Error.** We follow Gadre et al. [2024], predicting downstream average top-1 error (Err) across our benchmarks using the recorded validation loss (L), using a function of the form shown in Equation (2) where $\epsilon, k, \gamma$ are fit. In Figure 5, we fit a law to benchmark results sampled at every 10 billion

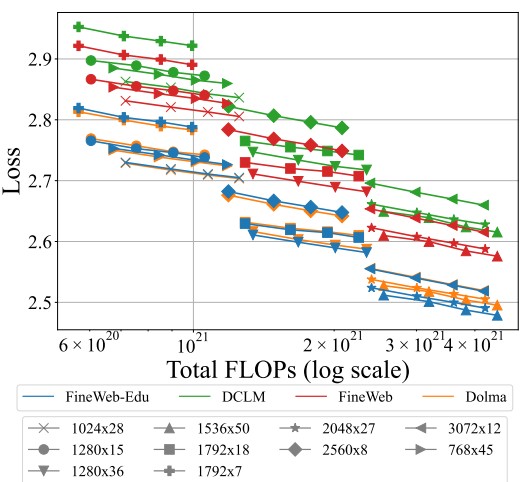

Figure 4: **Loss over multiple webtext datasets**. We see that the loss value changes for different datasets, including Dolma which we train on. DCLM and FineWeb have higher loss values whereas we measure lower loss values on FineWeb-Edu and Dolma. However, the rank order between models is stable across datasets. This suggests that it may be valid to fit scaling laws on various validation sets without necessarily needing to retrain the underlying models regardless of whether the validation data is i.i.d. with respect to the training distribution.

training tokens, we see that our fitted curve fits the data well. We observe greater variation around the fit compared to Gadre et al. [2024], which we attribute to the considerable differences in the width and depth of the Gemstones models.

**Predicting Benchmark Accuracy.** Following Bhagia et al. [2024], we calculate task loss by taking the loss over the correct answer to each benchmark question and averaging over all questions. We then use this task loss to predict average task accuracy across 4 downstream benchmarks. We find the accuracy of ARC, HellaSwag and MMLU to be most predictable at smaller compute scales and use this subset of benchmarks when fitting scaling laws to predict accuracy. Concurrently, Magnusson

---

[2]We note that there are 5 models in this subset for both Hoffmann et al. [2022] and our data, which meets the rule of thumb given by Choshen et al. [2024] for the minimum number of models that should be used to fit a scaling law.

et al. [2025] also observe this pattern across the same set of benchmarks. We predict average task accuracy by fitting a sigmoidal function of the form shown in Equation (3) where $a, b, k, l_0$ are fit. In Figure 6, we fit a law to benchmark results sampled at every 10 billion training tokens. We see a noisy fit and again suspect this is due to the variation in the Gemstones' width and depth. In Appendix E, we hold out the 2 billion parameter models and show extrapolation for both benchmark scaling laws and Approach 3 loss predictions.

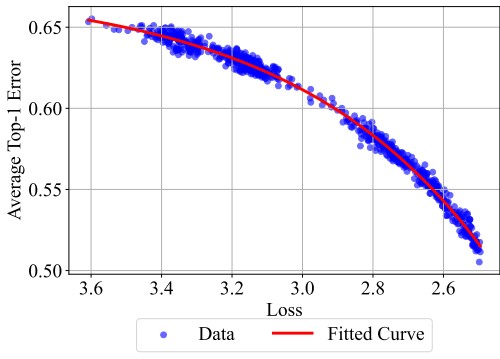

Figure 5: **Benchmark Scaling Law for Error.** We fit a law of the form shown in Equation (2) to benchmark results sampled at every 10 billion tokens and observe a tight fit.

Figure 6: **Benchmark Scaling Law for Accuracy.** We fit a law of the form shown in Equation (3) to benchmark results sampled at every 10 billion tokens for ARC, HellaSwag and MMLU.

$$\text{Err}(L) = \epsilon - k \cdot \exp(-\gamma L) \qquad (2)$$

$$\text{Acc}(L) = \frac{a}{1 + e^{-k(L - L_0)}} + b \qquad (3)$$

## 4.4 The Width/Depth vs. Compute/Time Continuum

In the previous section we show how deep models appear to achieve better final loss and accuracy on benchmarks when measured in terms of FLOPs. However, another crucial axis for practitioners to consider is the amount of wall time it takes to train a model. In Figure 7, we visualize in more detail the top of Figure 3, highlighting in color the approximate aspect ratio of the vertices that form our convex hull when fitting. On the left, we see that deep models are able to achieve a lower loss for a given computational budget (FLOPs) and therefore are selected as the vertices of our convex hull when fitting. However, on the right, we see that when the budget is measured in units of computer *time* (GPU hours), wider models become more pareto optimal. The concept of "overtraining" is an interesting dimension for further analysis, especially while varying width and depth, as one may be

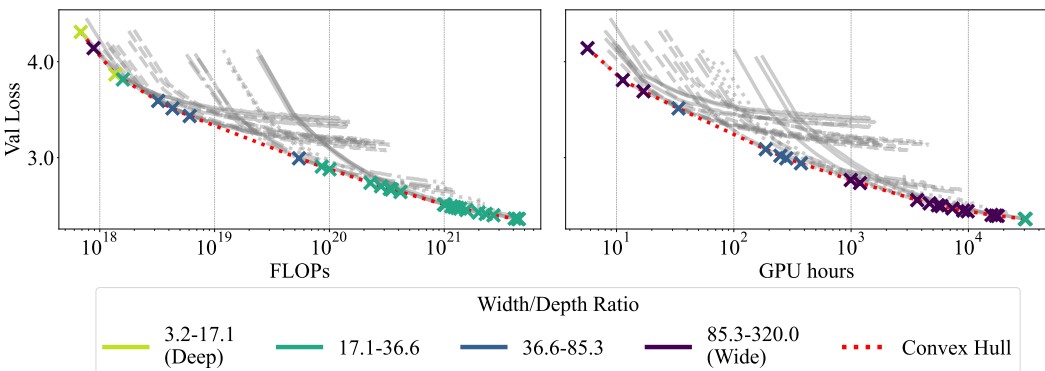

Figure 7: **Approach 1 fitting.** We enlarge and extend Figure 3 (top), highlighting in color the approximate aspect ratio of the vertices that form our convex hull over all 350 billion tokens of training. We see that wider models achieve a lower loss quicker in terms of GPU Hours (right) as the vertices of the convex hull are darker in color. However, deeper models (lighter in color) achieve a lower loss quicker in terms of FLOPs (left).

able to overtrain a smaller deep model to reach a low loss value quicker than a larger wider model. We analyze overtraining in greater detail in Appendix J.

We remark that while in Figure 7 the optimal models with respect to time tend to be the wider ones, this is probably due to our training scheme. Similar to other open-source efforts such as OLMo et al. [2024], we do not make any use of pipeline parallelism, and only employ tensor parallelism (using a hybrid data and tensor parallel algorithm similar to the ubiquitous Fully Sharded Data Parallel strategy). In summary, for standard parallelism implementations, wider models are simply easier to scale, but as a result our observations regarding resource overspending may not generalize to other parallelism strategies. As we open source all artifacts, practitioners can efficiently transform our open source results to suit their training setup. By simply running each model shape for only a handful of steps, recording the step times, updating the step time column in our fitting data and refitting the laws. This means that practitioners can easily transform our GPU Hours analysis to their specific hardware.

**Buried Treasure: Unearthing Value in Depth**    Finally, we plot the average benchmark accuracy (length normalized) of the Gemstones at 200, 250, 300 and 350 billion tokens. Figure 8 shows that the 1B scale models ($1280 \times 36$, $2560 \times 8$, and $1792 \times 18$) yield increasing accuracy with depth when constrained to approximately the same FLOP budget (vertically aligned points). We see similar patterns with the $768 \times 45$, $1280 \times 15$, $1792 \times 7$, and $1024 \times 28$ models, as well as the larger 2B models. This result is hinted at in Table 9 of the Gemma 2 report [Team et al., 2024b], where the authors note that for two models at the 9B scale the deeper model slightly outperforms the wider model across downstream benchmarks, but details of the exact experiment are sparse. Recent work suggests deeper layers in networks "do less" than shallower ones and can be pruned away [Gromov et al., 2024], but our downstream evaluations suggest that there are also advantages to additional model depth. We see similar patterns in the individual performance on each benchmark, and include those charts in Appendix Figure 16.

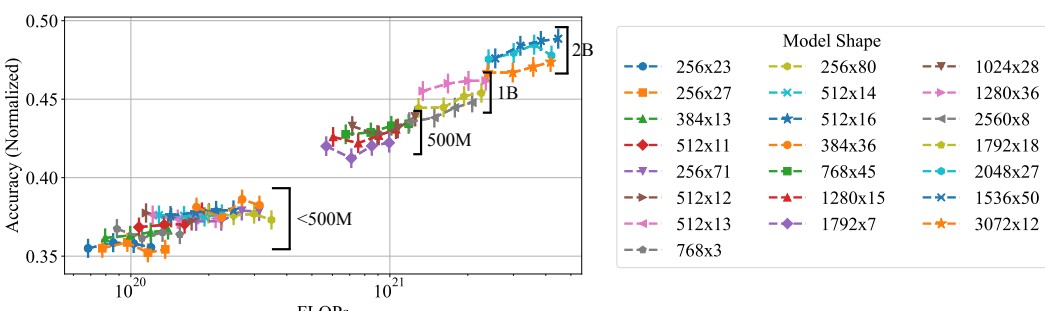

Figure 8: **Benchmark Performance.**    We benchmark all models using the 200, 250, 300 and 350 billion token checkpoints. Models show increasing accuracy with depth when constrained to approximately the same FLOP budget (vertically aligned points). This relationship between depth and accuracy can also be observed in many individual benchmarks (Figure 16).

## 5   Limitations and Conclusions

Altogether, our experiments and analysis demonstrate the impact of often overlooked design choices on scaling law outcomes, the importance of measuring the right type of performance metric, and the nuanced relationship between model width, model depth, computational budget, and training time. We hope this work encourages a rich range of future work based on the suite of open source artifacts we release. Potential avenues for extension include exploring hyperparameters that we kept constant such as the expansion factor of the transformer (the ratio by which the dimensionality of the hidden layer in feed-forward network increased relative to its input dimension), the vocabulary size, the learning rate schedule, and the batch size. Although we endeavor to make our laws as generalizable as possible, we still expect that their applicability declines in training set-ups very different from our own.

## Acknowledgments and Disclosure of Funding

We thank (in alphabetical order): Brian Bartoldson, Bhavya Kailkhura, Avi Schwarzschild, Sachin Shah and Abhimanyu Hans for helpful feedback.

An award for computer time was provided by the U.S. Department of Energy's (DOE) Innovative and Novel Computational Impact on Theory and Experiment (INCITE) Program. This research used resources of the Oak Ridge Leadership Computing Facility at the Oak Ridge National Laboratory, which is supported by the Office of Science of the U.S. Department of Energy under Contract No. DE-AC05-00OR22725.

This work was made possible by the ONR MURI program, DAPRA TIAMAT, the National Science Foundation (IIS-2212182), and the NSF TRAILS Institute (2229885). Commercial support was provided by Capital One Bank, the Amazon Research Award program, and Open Philanthropy.

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

# A  Model Implementation Details

All models have a head size of 128 because 256 is the maximum head dimension supported by the AMD implementation of Flash Attention 2 we utilize and we constrain our search to models with $> 1$ attention heads. We assume the simple convention of the Llama series where the head dimension is always the embedding dimension divided by the number of heads, implying that the embedding dimension (width) must be divisible by 128. Following conventions from the Gemma suite, we constrain the head count to be even to enable Grouped Query Attention [Ainslie et al., 2023] with a query to key ratio of $2 : 1$ and we fix the intermediate size to be $4\times$ the width of the model. We choose our vocabulary size to match the $50,304$ tokens in the Pythia tokenizer. While many of the architecture choices mirror those from Gemma, for simplicity we do not use logit softcapping nor do we tie the embedding and language modeling head weight matrices.

## A.1  Optimal Learning Rates for Gemstones

Training models across diverse shapes and scales requires learning rates that ensure both stability and near-optimal performance. Suboptimal learning rates risk misrepresenting scaling laws, as they could conflate architectural preferences with hyperparameter sensitivity. For the Gemstone models—varying in width, depth, and size—we address this challenge through a unified learning rate scaling rule and a parameter initialization scheme tailored for stability.

**Unified Learning Rate Scaling Rule**  Existing scaling rules prescribe learning rates ($lr$) as $lr_{\text{base}}/\text{width}$ for width scaling or $lr_{\text{base}}/\sqrt{\text{depth}}$ for depth scaling. Since Gemstone models vary both dimensions, we propose a hybrid rule: $lr_{\text{eff}} = lr_{\text{base}}/(\text{width} \times \sqrt{\text{depth}})$ This accounts for the compounding effect of gradient dynamics across width and depth, balancing update magnitudes during optimization.

**Empirical Validation**  To validate $lr_{\text{base}}$, we stress-test four extreme model shapes: wide (64 layers, 768 width) and deep (128 layers, 512 width) at 100M and 2B parameter scales. Each is trained for $1k$ steps with a batch size of 2048 and context length of 2048 (4.2B tokens). We sweep $lr_{\text{eff}}$ from $10^{-4}$ to $5 \times 10^{-2}$. As shown in Figure 9 (left), optimal $lr_{\text{eff}}$ varies widely across architectural shape. However, rescaling the x-axis by width $\times \sqrt{\text{depth}}$ collapses all curves onto a shared trend, revealing $lr_{\text{base}} = 5$ as the consistent optimum (right panel). This confirms our rule's efficacy for width-depth transfer.

**Flaws in the Gemstones.**  While $lr_{\text{base}} = 5$ achieves stable training for most models under the scheme described above, wider architectures (e.g., 256 width-depth ratio) occasionally exhibit loss spikes nonetheless. Despite these instabilities, via rollbacks and minor modifications to the learning rates for the most extreme models, all models in the suite are trained to 350B tokens without divergence. We discuss these issues and our solutions further in Appendix K.2.

**Ablation Study**  To assess sensitivity to $lr_{\text{base}}$, we replicate training for a subset of models with $lr_{\text{base}} = 2.5$ (e.g. dividing $lr_{\text{eff}}$ by 2). While losses are marginally higher, scaling law fits remain robust, suggesting our conclusions are not artifacts of aggressive learning rates.

**Scalable Parameter Initialization Rules.**  Finally, stable training across model shapes and scales also requires model specific tweaks to parameter initialization Yang et al. [2021]. Following OLMo(1) [Groeneveld et al., 2024], we apply a parameter initialization strategy intended to enable stable training and learning rate transfer across scales. We initialize all parameters as truncated normal ($\mu = 0, a = -3 \cdot \sigma, b = 3 \cdot \sigma$) with modified variances dependent on the parameter type. We use $\sigma = 1/\sqrt{\text{width}}$ except for the attention projections which are initialized as $\sigma = 1/\sqrt{2 \cdot \text{width} \cdot (l+1)}$ and the MLP projections as $\sigma = 1/\sqrt{2 \cdot (4 \times \text{width}) \cdot (l+1)}$ where in each case $l$ is the layer index (not the total model depth) and the $4\times$ factor comes from the relation of width to MLP intermediate dimension.

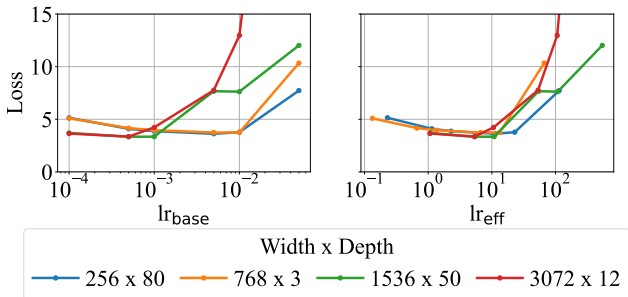

Figure 9: **Learning rate scaling is necessary for width-depth transfer.** Left: Preliminary training runs with initialization rules active, but no learning rate scaling. Right: Same data, but with x-axis rescaled to simulate the application of learning rate scaling with $lr_{\text{base}} = lr_{\text{eff}} \times (\text{width} \times \sqrt{\text{depth}})$.

## A.2  Software and Data

We train all models using a fork of `litgpt` [AI, 2023] enhanced with AxoNN [Singh and Bhatele, 2022, Singh et al., 2024] tensor parallelism. We open source all models used in our analysis to Hugging Face [Wolf et al., 2020] and the logging from training on Weights and Biases in json format.

## B  Extended Related Works

Scaling laws have been constructed in many different areas of machine learning since their original proposal in [Kaplan et al., 2020]. Early work on scaling laws for machine translation by Ghorbani et al. [2021] splits the parameters term into two, one for each of encoder and decoder components, and similarly to Gordon et al. [2021] analyzes the relationship between BLEU scores and scaling laws. Subsequently, Zhang et al. [2022] and Bansal et al. [2022] studied the impact of architecture choice on the scaling law, finding increasing data or parameters can compensate for worse architectural decisions. However, the advent of performant open source pipelines for language model development following the release of the Llama series [Touvron et al., 2023a] spurred a renewed flurry of interest in the topic in academic settings.

**Architecture**  Allen-Zhu and Li [2024] builds scaling laws to model how specific architectural choices impact measures of knowledge acquisition (bits-per-param) in highly controlled settings; they run parallel experiments across dimensions like architecture, quantization, and sparsity to derive insights about which design choices affect acquisition and storage capacity the most. However the more general study of scaling laws for sparse architectures is quite extensive. Clark et al. [2022] analyze how the number of experts can be used in the law, studying both linear and quadratic interactions for many types of routing models. Frantar et al. [2023] focus on weight sparsity within foundation models, adding a multiplicative parameter on the parameters term in the law. Yun et al. [2024] analyzes the trade offs between optimal training and optimal inference and Krajewski et al. [2024] find that with optimal settings, a Mixture of Experts model always outperforms a transformer model at any computational budget. Model shape has also been analyzed for sparse mixture of expert models and in the context of finetuning. Krajewski et al. [2024] use the ratio between the standard feed-forward hidden dimension and the hidden dimension of an individual expert to allow their law for mixture of expert models to predict the width of the experts.

**Downstream Benchmarks**  Beyond modeling just training loss—the canonical prediction target for scaling laws—there are multiple works analyzing whether scaling laws can be used to predict performance on downstream tasks. Ruan et al. [2024] show that scaling laws can be predictive of benchmark performance. Caballero et al. [2023] observe that traditional scaling laws cannot capture complex behaviors like non-monotonic trends nor inflection points. They propose broken scaling laws–a piecewise or smoothly broken power law–that better predicts performance of both upstream and downstream tasks.

**Finetuning and Data-constrained Regimes**  Further analyses using scaling laws have extended to analyzing finetuning and data limited scaling. Hernandez et al. [2021] find that finetuning is much

more compute efficient when the pretraining ignored. Zhang et al. [2024] study parameter efficient finetuning regimes find a multiplicative law is better for the finetuning setting than the classical additive law used by others. Muennighoff et al. [2023] analyze the data constrained training regimes, finding epoching data up to four times is as good as training on deduplicated data in terms of reducing loss.

**Multi-modality**   These techniques are not limited to generative text modeling only; they have also been applied to multi-model models. Henighan et al. [2020] find optimal model size can be described as a power law for model modeling including images and video. The authors also find that model size does not help 'strong generalization' for problem solving. Aghajanyan et al. [2023] analyze text, images, code and speech, presenting a scaling law to describe the competition between these modalities and describe a regime for optimal hyperparameter transfer from the unimodal to multimodal regimes. Liang et al. [2024] look at scaling laws for diffusion transformer models. Li et al. [2024a] analyze scaling laws for vision encoder commonly used to encode image inputs for transformer model backbones, finding increasing the size of the encoder alone can lead to performance degradation in some cases.

**Zero-shot Hyperparameter Transfer**   The ability to train a series of models with extremely different parameter counts is an implicit requirement of any scaling law analysis. Bjorck et al. [2024] find that optimal learning rates change with length. Work on zero-shot hyperparameter transfer across transformer model *widths* is mature [Yang et al., 2021, Everett et al., 2024, Hayou and Yang, 2023, Dey et al., 2024]. Achieving transfer across diverse model *depths* is less well studied, especially in transformer language models [Bordelon et al., 2024, Yang and Hu, 2021, Yang et al., 2023]. While Yang et al. [2023] argue that depth transfer requires scaling in $1/\sqrt{L}$ because this is the unique regime with maximum feature diversity, more recently Dey et al. [2025] argue that instead the scaling should be in $1/L$.

### B.1   Chinchilla Equation 4

Hoffmann et al. [2022] use the variable names $N$ and $D$ for the number of parameters and number of tokens respectively, defining their parameterized form as:

$$\hat{L}(N, D) \triangleq E + \frac{A}{N^\alpha} + \frac{B}{D^\beta}, \tag{4}$$

Equation 4 of Hoffmann et al. [2022] is defined as:

$$N_{opt}(C) = G\left(\frac{C}{6}\right)^a, \quad D_{opt}(C) = G^{-1}\left(\frac{C}{6}\right)^b, \tag{5}$$

$$\text{where} \quad G = \left(\frac{\alpha A}{\beta B}\right)^{\frac{1}{\alpha+\beta}}, \quad a = \frac{\beta}{\alpha + \beta}, \text{ and } b = \frac{\alpha}{\alpha + \beta}. \tag{6}$$

## C   Data Sampling

We plot the entire space of all possible models subject to our design constraints discussed in Figure 10. While exploring the impact of finer grained depth differences during our experiments, we decided to add two additional models slightly outside the $\pm 5\%$ tolerance band at the 100M scale; for $width = 512$, in addition to the originally chosen depths of 12 and 13, we added 11 and 14; these appear as a dense collection of 4 points at the same $width$.

## D   Mathematical Definition of Our Convex Hull Method

  We give a mathematical definition for our convex hull method, loosely based on the wikipedia entry for reconstructing functions from epigraphs:

We can define the set of points we have to fit on as FLOPs/GPU hours $(x)$, loss value $(L)$ pairs.

$$\mathcal{D} = \{(x_i, L_i)\}_{i=1}^n \subset \mathbb{R}^2$$

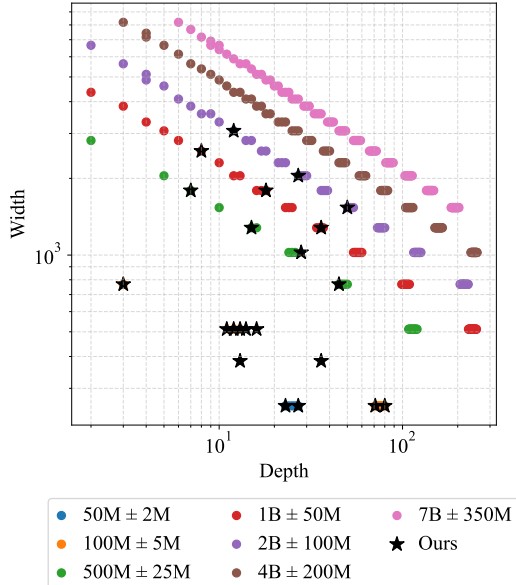

Figure 10: All possible model shapes we could have chosen based on our architecture within $\pm 5\%$ are shown as circles. The points we selected are highlighted as stars, including the two extra points we select to have four models of width $512$.

Let $\mathrm{conv}(\mathcal{D})$ denote the convex hull, the linear interpolation of any two points in $\mathcal{D}$:

$$\mathrm{conv}(\mathcal{D}) = \left\{ \sum_{i=1}^{n} \lambda_i (x_i, L_i) \ \middle|\ \lambda_i \geq 0,\ \sum_{i=1}^{n} \lambda_i = 1 \right\}$$

Define

$$\hat{L}(x) = \min(\{y | (x, y) \in \mathcal{D}\})$$

The lower convex hull is the graph of this new function:

$$\{(x, \hat{L}(x)) | x \in \mathrm{Dom}(\hat{L})\}$$

We think the easiest visualization of this is in Figure 7 where the red line is the convex hull, the colored crosses are the vertices and the gray lines are all possible points in the dataset.

## E  Extrapolation to Larger Models

To quantify the robustness of our scaling laws to changes in model scale, we perform two types of extrapolation analyses. In the first experiment, we hold out the "$2B$" parameter models, fit scaling laws to the samller models, and then test the extrapolation performance of the fitted law. Due to the $10\%$ tolerance margin in our parameter count strata, in actuality we fit benchmark scaling laws using models with less than $1.8B$ parameters and extrapolate to models with more than $1.8B$ parameters. In Figure 11 we see that extrapolating to predict error offers a much better fit than we see in Figure 12 when predicting accuracy.

In a second type of experiment, we again analyze extrapolation as a function of model size but this time quantifying prescription robustness in terms of estimated versus actual validation loss (using approach 3). Following Choshen et al. [2024], we report the mean absolute relative error (ARE) over the last $30\%$ of training tokens ($250b$ to $350b$ tokens for Gemstone models). We find the ARE when fitting on all checkpoints and then testing on models with more than $1.8B$ parameters is $0.68\%$. When only fitting on models with less than $1.8B$ parameters and extrapolating to models with more

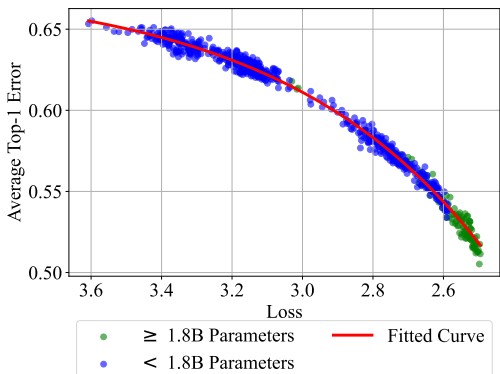

Figure 11: **Benchmark Scaling Law for Error.** We fit a law of the form shown in Equation (2) to benchmark results sampled at every 10 billion tokens using models with less than $1.8B$ parameters and observe a tight fit when extrapolating to models with more than $1.8B$ parameters.

Figure 12: **Benchmark Scaling Law for Accuracy.** We fit a law of the form shown in Equation (3) to benchmark results sampled at every 10 billion tokens for ARC, HellaSwag and MMLU using models with less than $1.8B$ parameters and observe a poor fit when extrapolating to models with more than $1.8B$ parameters.

than $1.8B$ parameters, the ARE is $0.63\%$. Choshen et al. [2024] find ARE's of up to $4\%$ are typically used to distinguish between modeling choices, implying that our extrapolation error is well within the acceptable range.

# F   Leave-One-model-Out Analysis

To evaluate how robust our scaling laws are to a different aspect of our experimental design, we estimate the variability in our fitting process caused by our model selection using a leave-one-out analysis. In Figure 13 we re-fit the same scaling law multiple times leaving each one of the model shapes out in turn using both Approach 1 and Approach 3. We visualize these results by plotting the minimum and maximum tokens per parameter ratios yielded across all leave-one-out trials along with the prescription based on all data (bounds of the gray shaded region vs the green line). While the implications of the precise min/max values are somewhat up to interpretation, compared to the difference between our fit, Kaplan's, and Chinchilla's, the relative narrowness of the gray region suggests that any disagreement in tokens per parameter prescription we find versus prior work is unlikely to simply be an artifact of (our) specific model selection. Finally, comparing the left and right sides of Figure 13, the smaller gray region in the former suggests that Approach 1 is less sensitive to this model re-sampling process than Approach 3. We hypothesize that at least some of this increased robustness in fit can be attributed to our novel variance-reducing convex hull method applied in Approach 1.

# G   Variability in fitting

In Table 2, we show a similar table to Table 1 but exclude embeddings from the parameter count in the fitting process. We also visualize all fitted lines in Figure 14.

## G.1   Variability from sampling

In this section, we visualize the variability in the fitting process due to the frequency of sampling the data. We do this by sampling fitting data every 2 billion tokens of training and every 10 billion tokens of training and comparing the laws found. In Figure 15, comparing the "Ours" lines, we see that for Approach 3 the difference is minimal where as for Approach 1 there is a change in the law. This can be intuitively explained as the data on the hull is much sparser the points the law is fitted to

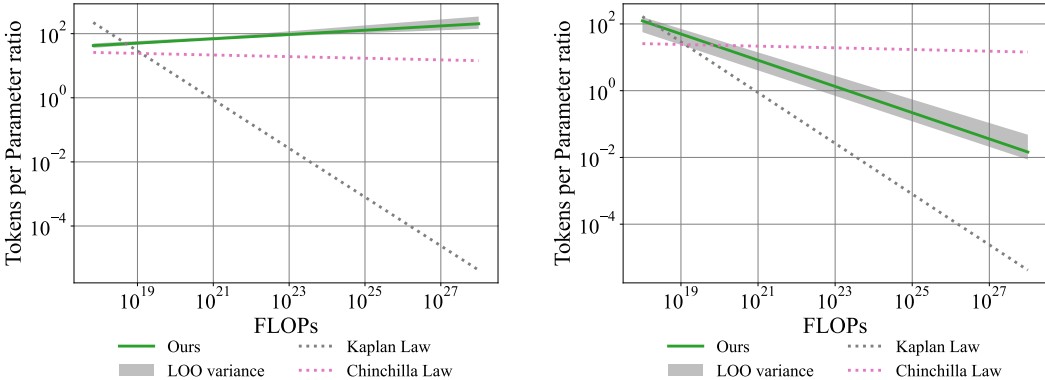

Figure 13: **Leave-One-Out Validation.** We leave each model out and refit the law to obtain the gray shaded region (min/max ratio across all trials) and compare it with the law fit to all data in green. **Left)** shows the result when Approach 1 is used, and **right)** shows the result of using Approach 3.

Table 2: **We demonstrate the variability in fitting scaling laws by resampling our data many different ways.** The slope can be viewed as the exponent in the power law relationship $parameters = constant \cdot compute^{exponent}$. Grouping by fitting approach and choice to include embeddings, in the final column 'Delta' we show the change in slope produced by the ablations against the corresponding base law fit on the full set of hot data. Values with an absolute magnitude greater than 0.05 are highlighted in orange, and those exceeding 0.1 are highlighted in red. We see that the reduced sampling has a large impact on the slope of the law and that Approach 1 is more sensitive than Approach 3. We plot these prescriptions in Figure 14 and show this table with embeddings included in the parameter count in Table 1.

| Tokens | Cooldown | LR Ablation | Embeddings | Slope | Delta |
|---|---|---|---|---|---|
| Kaplan et al. [2020] | | | | 0.7300 | |
| **Approach 1 (no Embeds)** | | | | | |
| all | ✗ | ✗ | ✗ | 0.5689 | |
| $\leq 100b$ | ✗ | ✗ | ✗ | 0.6269 | 0.0579 |
| $> 120b$ | ✗ | ✗ | ✗ | 0.9666 | 0.3977 |
| all | ✗ | ✓ | ✗ | 0.6224 | 0.0535 |
| all | ✓ | ✗ | ✗ | 0.7242 | 0.1552 |
| **Approach 3 (no Embeds)** | | | | | |
| all | ✗ | ✗ | ✗ | 0.7141 | |
| $\leq 100b$ | ✗ | ✗ | ✗ | 0.7030 | -0.0111 |
| $> 120b$ | ✗ | ✗ | ✗ | 0.7350 | 0.0209 |
| all | ✗ | ✓ | ✗ | 0.6929 | -0.0211 |
| all | ✓ | ✗ | ✗ | 0.7104 | -0.0037 |

changed, hence for Approach 1 more fitting data gives a more reliable fit. Hence, in all other plots in this paper for Approach 1 we fit with data recorded every 2 billion tokens for accuracy and for Approach 3 we fit with data every 10 billion tokens for speed.

Further, in Figure 15, we also present laws fitted on the DCLM and FineWeb-Edu data in Figure 4. For the DCLM and FineWeb-Edu data, we record data every 10 billion tokens of training. In Figure 15, we see that difference between the laws found by fitting on different data sets compared to our main analysis on Dolma is minimal.

# H    Individual Benchmark Results

In Figure 16, we see zero-shot MMLU scores of our larger models are quite non-trivial at $28\% - 33\%$, despite being trained on an open dataset, without a cooldown period, or any sort of post-training.

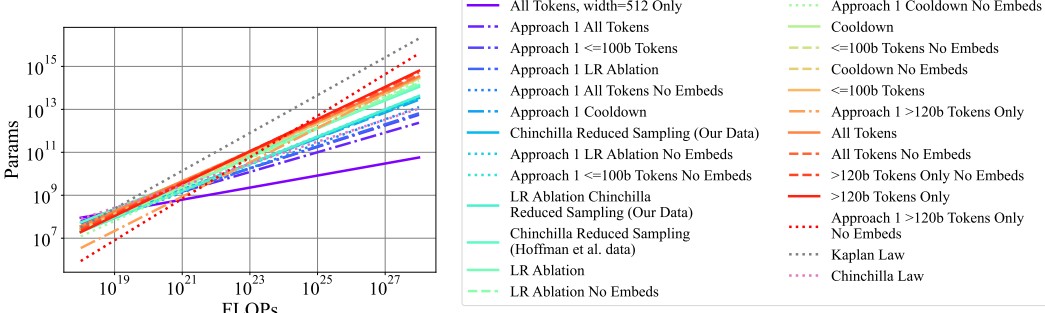

Figure 14: **We demonstrate the variability in fitting scaling laws by resampling our data many different ways.** We label prescriptions found using Approach 1 with "Approach 1" in the legend, otherwise approach 3 is used. All tokens counts available are used to fit the laws unless stated otherwise in the legend, for example $\leq 100B$ means that only token counts less than or equal to $100B$ are used in fitting.

*No Embeds*: Embedding parameters are not counted when fitting these laws.

*Cooldown*: Only data from the cooldown ablation is used to fit this law.

*LR Ablation*: Only data from the learning rate ablation training runs, where the learning rate is halved, is used to fit these laws.

*width=512 Only*: Only models with width 512 are used to fit these laws.

*Chinchilla Reduced Sampling*: We subsample our data to be as close as possible to the token counts and model sizes that Hoffmann et al. [2022] use to fit their scaling laws and also fit new scaling laws on this subset of Hoffmann et al. [2022] data. Details in Section 4.2.

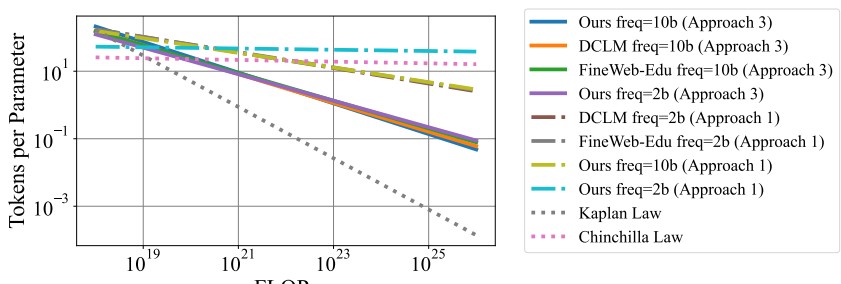

Figure 15: We fit laws when data is sampled every 2 and 10 billion tokens of training. We also compare to laws fit on DCLM and FineWeb-Edu data. We see sampling frequency is important for Approach 1 and that the difference laws fitted on FineWeb-Edu, DCLM, and Dolma is small.

# I  The Price of Stepping Off the Scaling Law

By analyzing the cost of stepping off of the scaling law, we find that some kinds of design errors are more damaging than others. We also find that training on more tokens than is strictly recommended (aka "overtraining") is typically quite efficient in terms of pushing down loss.

**If You Value Your Time, Train Wide Models.** We first show that in our training setup, training wider models is far more efficient than training deep models. In Figure 17, we reflect on the consequences of suboptimal architectural choices, by considering how much of a given resource—FLOPs or GPU hours—would be "overspent" to reach any target loss value with the plotted architecture rather than the prescribed width and depth. We find that choosing to train "skinny" models (top left) wastes many FLOPs and GPU hours. The scale of overspend is quite different however, with the least efficient models only overspending about $50\%$ on FLOPs but wasting more than $200\%$ of the GPU hours spent by the best configuration. In other words, in the time taken to train a single (very) suboptimal model to the desired loss value, one could train three optimal-width-depth models. We note that while the time-optimal models tend to be the wider ones, this is probably due to our training scheme. Similar to other open-source efforts such as OLMo et al. [2024], we do not make any use of pipeline

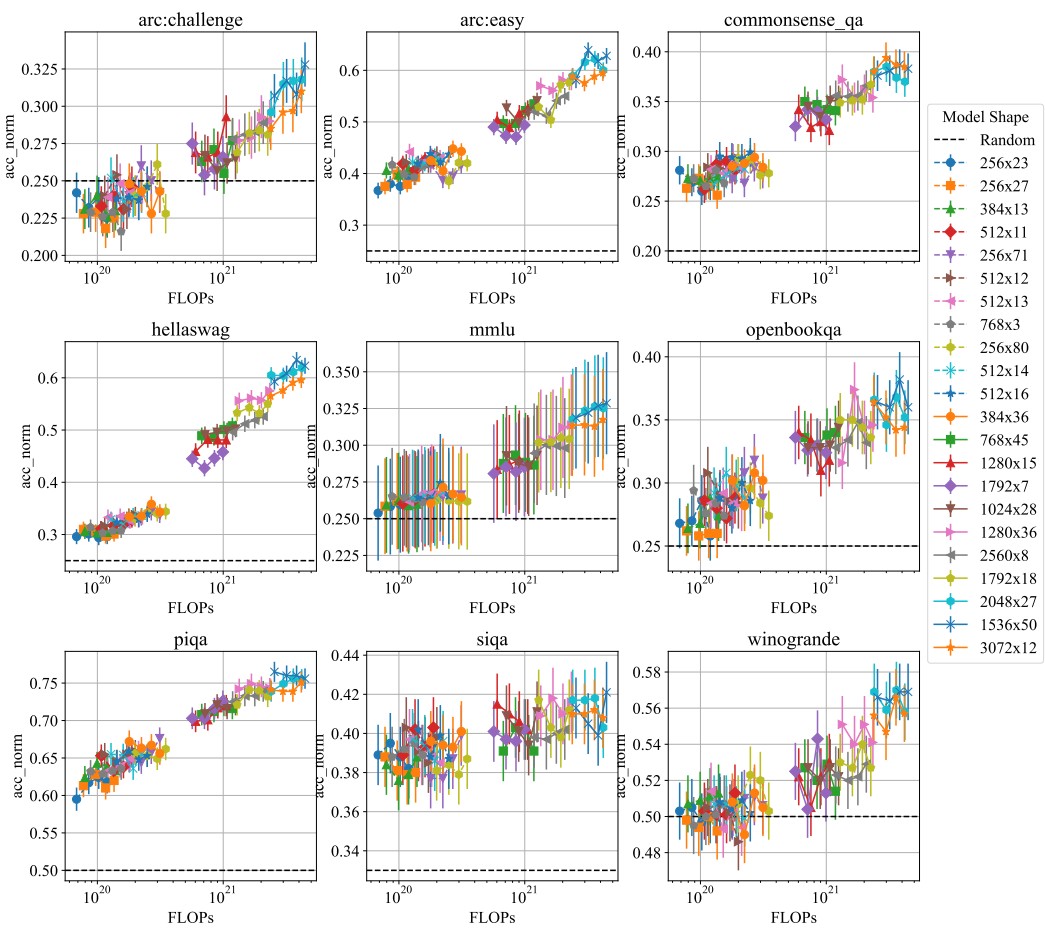

Figure 16: **Individual benchmark performance.** We benchmark all models using the 200, 250, 300, and 350B token checkpoints, converting this to FLOPs using the formula shown in Appendix M. Models show increasing accuracy with depth when constrained to approximately the same FLOP budget (vertically aligned points) on many benchmarks. We plot the average benchmark accuracy in Figure 8.

parallelism. In summary, for standard parallelism implementations, wider models are simply easier to scale, but as a result our observations regarding resource overspending may not generalize to other parallelism strategies.

## J    Scaling Laws Predict That Overtraining Is Efficient.

Similarly to Gadre et al. [2024], we can shift optimal points to simulate overtraining. To do this, we fix a FLOP budget and trace out a path of model sizes and corresponding token counts to remain within that budget. For each model size and token count, we record the "overtraining factor," which is the selected number of training tokens divided by the optimal number of tokens for that model shape. An overtraining factor of less than one corresponds to undertraining the model, and a factor greater than one represents overtraining. We show the results of this process in Figure 18. We see that overtraining does increase predicted loss at a given FLOP count but that these curves are actually quite flat. We include the loss values of open source models on our own validation set to allow readers to contextualize the y-axis values. Especially at high FLOP counts, our laws predict overtraining becomes quite efficient in that it results in fairly small elevations in loss for a relatively large reduction in model size.

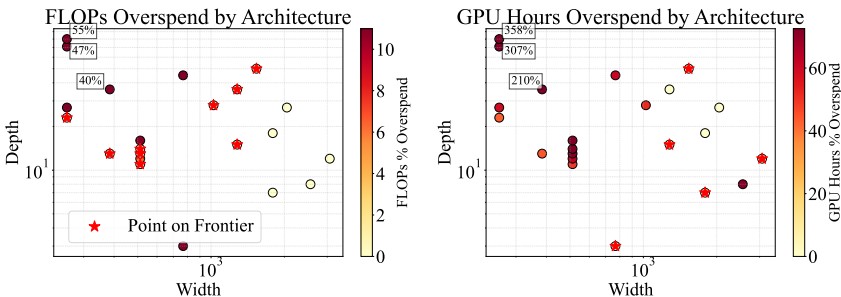

Figure 17: **The inefficiency of training models with suboptimal widths and depths.** We plot the FLOPs (left) and GPU Hours (right) *overspend* after training our Gemstones for 300 billion tokens. We define the overspend as how many resources (FLOPs or GPU hours) are required for a model with a given width-depth configuration to reach some target loss, relative to the models that achieve that target loss the fastest (the "points on (pareto)-frontier"). We can see that the skinny models (top-left, dark points) use many more FLOPs or GPU hours to reach a target loss than the wide models. We note that these inefficiencies exist in our training setup because we only use tensor parallelism and not pipeline parallelism but highlight how to easily transfer these results to other environments in Section 4.4.

Industry models often use fewer parameters and train on more tokens than prescribed in prior work. We find the impact of overtraining a smaller model on predicted loss to be small. Combining this with Figure 17, where wider models are predicted to be optimal in terms of GPU hours, reinforces the message that FLOPs optimality is not the end of the story for training models. Trading some FLOPs optimality for time optimality necessarily means overtraining, but Figure 18 suggests the difference is marginal. We believe this combined evidence makes significant progress towards explaining the differences between the prescriptions found in prior work and training choices observed in industry.

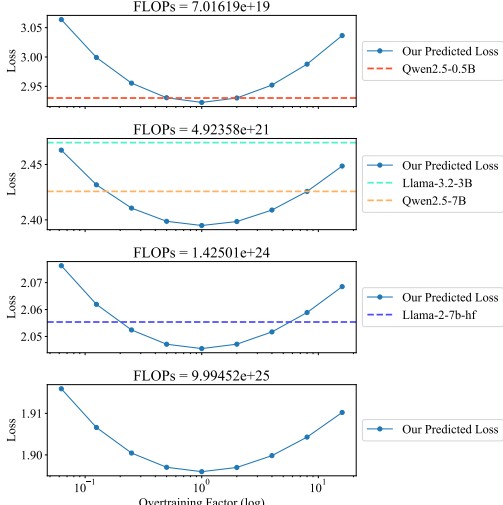

Figure 18: **Quantifying the cost of overtraining.** We simulate deviations from our prescriptions to assess their impact on model performance by increasing the optimal token count prescribed by an overtraining factor. We then optimize the model shape to achieve the lowest loss possible at each FLOP budget and overtraining factor. Note that $10^0$, or $1\times$, is the prescribed optimal point. We take four FLOP budgets (title of each plot) and plot the loss as a function of overtraining factor and see that under or overtraining increases predicted loss but by only a small amount. We plot the losses of selected open source models on our validation set to help ground the y-axis ranges.

## K   Training

Despite our best efforts to sufficiently mix the training data, we still see slight jumps in the global training loss when the training switches between chunks of data, hence we use validation loss to fit all laws as this is smooth.

### K.1   Loss Curves

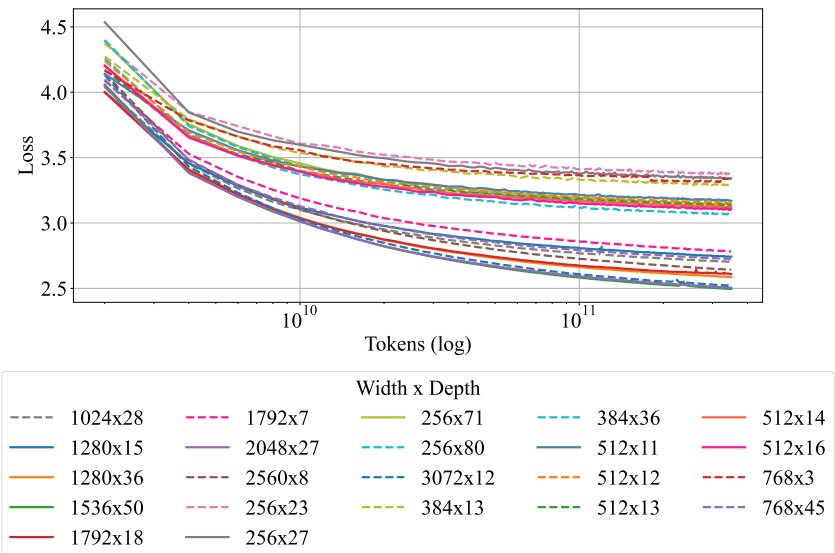

Figure 19: Loss curves for the main 22 training runs.

### K.2   Additional Training Complications

Any gemstone naturally contains a small number of inclusions or fractures. We discuss a few of the minor imperfections in our model collection below.

**Dealing with Training Instabilities**   After some of the widest models were trained beyond $50B$ tokens we began to observe unrecoverable loss spikes that were proceeded by small wobbles in the loss trajectory. Under the general intuition that the culprit was most likely that the $width/depth$ ratios considered were simply *too extreme* for existing initialization and learning rate scaling approaches to handle, we reran some of the models with a "patch" in place.

We modified the initialization rules and learning rate scaling factors to rescale the depth and layer indices of the model such that if $width/depth > 256$ scale variances and learning rates as if the depth of the model was actually $depth' = \lceil (width/100) \rceil$. The overall effect of the patch is to initialize and scale learning rates more conservatively, as if the aspect ratio were only 100 while keeping the original $width$ of the model. We found this allowed us to complete training for a full set of 22 models out to 350B tokens for even our most extreme models.

However, after $350B$ tokens, despite these efforts we observed that most extreme models which were patched still diverged anyway. While a partial cause of this could be the constant learning rate scheduler employed during training, concurrent work, from the authors of the original OLMo paper and codebase [Groeneveld et al., 2024] from which we derived some of our choices, reported that the initialization scheme dubbed the "Mitchell-init" is indeed systematically prone to instabilities later on in training [OLMo et al., 2024]. While an unfortunate finding, we were unable to rerun all of our experiments due to the consumption of significant non-fungible compute resources in the original experiments.

**Models Lacking Ablations**   Our cooldown ablation is from initial experiments below $100B$ tokens of training which do not use the patched learning rates scaling rules. This means there are minor

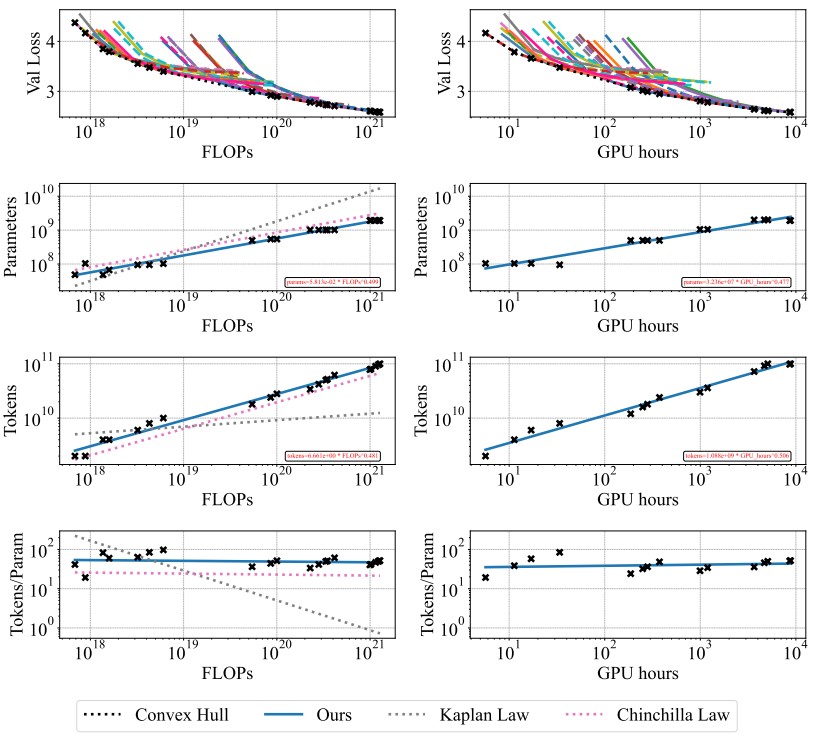

Figure 20: Extended Approach 1 plot from Figure 3, including tokens and parameters axes. As in Figure 3, we present an analysis over FLOPs on the left and over GPU hours take to train on the right.

discrepancies between the cooldown ablation and main set of training runs for the widest models from the three largest parameter count groups ($1792 \times 7$, $2560 \times 8$, $3072 \times 12$). We also do not cool down the $100B$ token checkpoint for the $3072 \times 12$ model, as it was experiencing a loss spike at that final point. Finally, we do not include ablations for the two width 512 models which do not fall into the $\pm 5\%$ boundary of the $100M$ parameter count ($512 \times 11$, $512 \times 14$), as they were only added to the collection in later experiments.

# L   Ablations for Approach 1

## L.1   Extended Paper Figures

In Figure 20, we plot an extended version of the Approach 1 plot we present in Figure 3.

## L.2   Alternative Learning Rates

In Figure 21, we present the Approach 1 prescription when fitting on the learning rate ablation data.

## L.3   Cooldown

In Figure 22, we present the Approach 1 prescription when fitting on the cooldown ablation data.

## L.4   Varying Delta in the Huber loss

In Figure 23, where we plot the exponents found by optimizing the Huber loss versus the size of the grid search used for optimization. We see that a delta of $10^{-5}$ is unstable for smaller grid sizes and

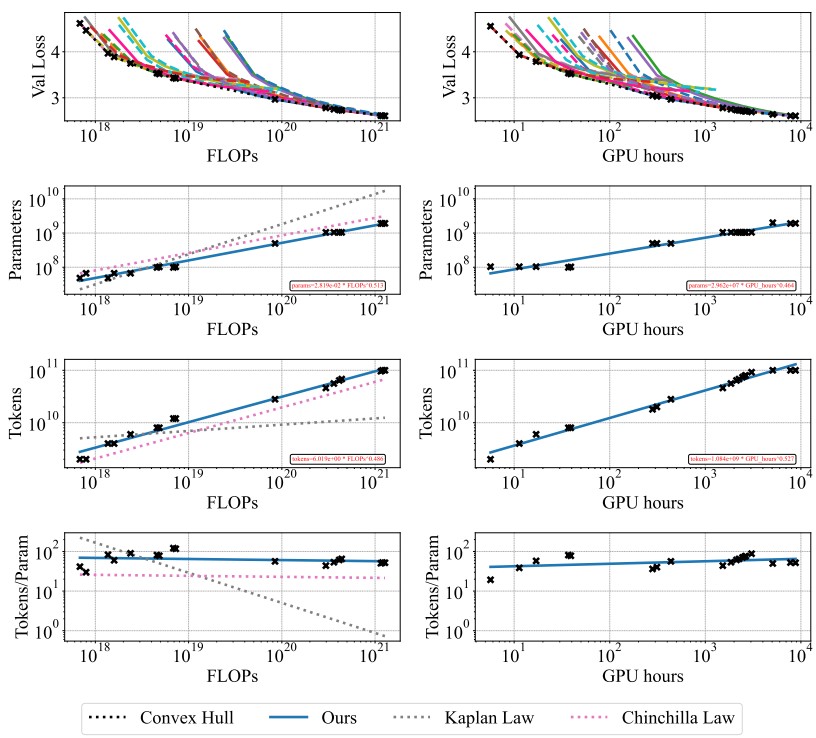

Figure 21: Approach 1 fitted on the learning rate ablation dataset. As in Figure 3, we present an analysis over FLOPs on the left and over GPU hours take to train on the right.

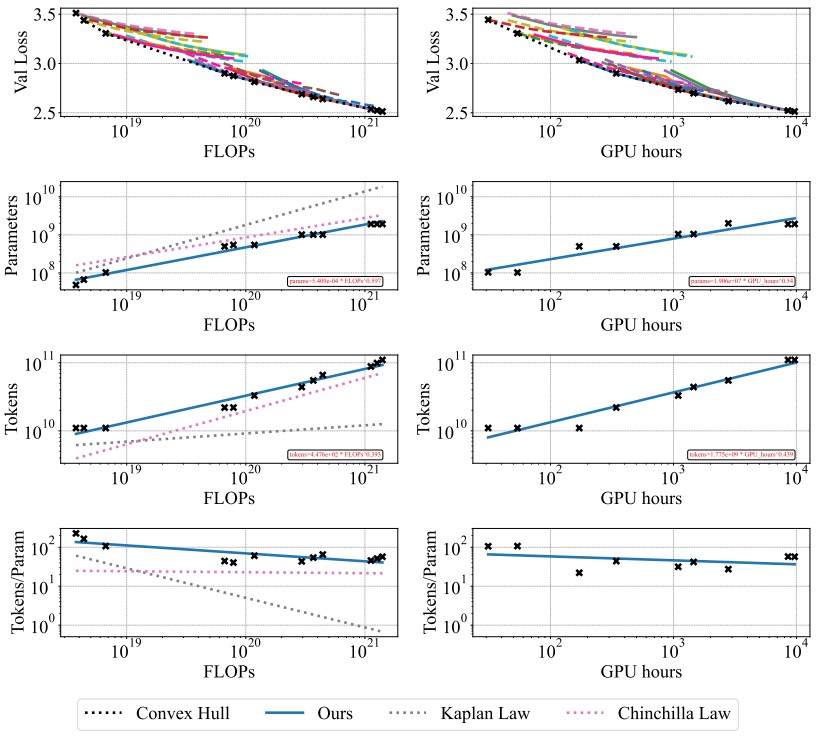

Figure 22: Approach 1 fitted on the cooldown ablation dataset. As in Figure 3, we present an analysis over FLOPs on the left and over GPU hours take to train on the right.

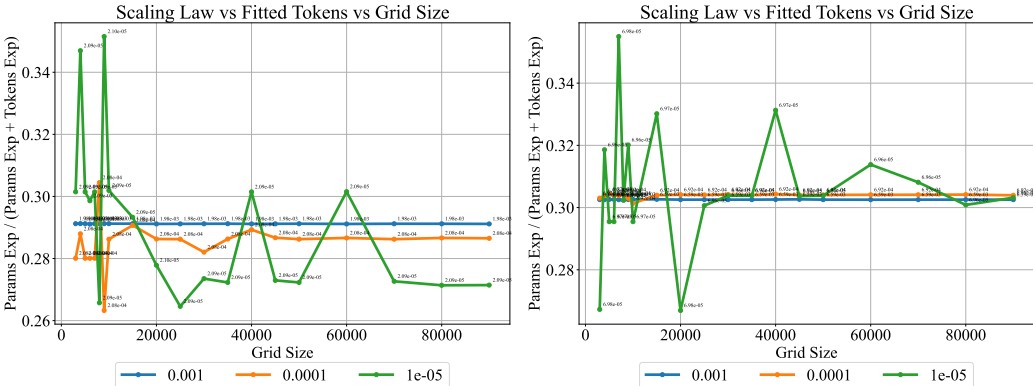

Figure 23: We plot the size of the grid search as the x axis and the gradient of the prescribed tokens as the y axis. We vary delta and see that a delta of $10^{-5}$ is highly unstable when fitting on smaller grid sizes. On the left, we plot only fitting on data less than 100 billion tokens. On the right, we plot fitting on all data up to 350 billion tokens. We see that including more data increases the stability of the exponents found for smaller grid sizes for deltas $10^{-4}, 10^{-5}$.

including more tokens in the fitting data generally increases stability of the exponents found during optimization.

## M    FLOP counting matters

In Figure 24 we show that the common approximation of FLOPs per token$= 6 \times parameters$, miscounts the true FLOPS by a significant amount.

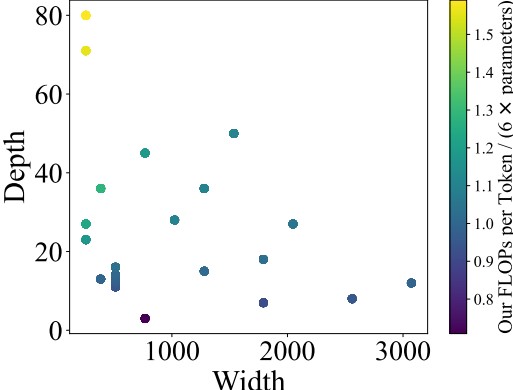

Figure 24: We color the points based on the ratio of our calculated FLOPs per token which is shown in the code below and using $6 \times parameters$. We see counting the FLOPs properly becomes more important for aspect ratios off outside of the standard regime.

```
VOCAB_OURS = 50304
SEQ_LEN = 2048
WORLD_BATCH_SIZE = 2048.0
HEAD_SIZE = 128
EXPAND_FACTOR = 4.0

def flops_per_token_gqa(
    width: NDArray[number] | number,
    depth: NDArray[number] | number,
    vocab_size=VOCAB_OURS,
```

```python
    queries_per_group=2,
    seq_len=SEQ_LEN,
):
    """
    Some details (negligible even for extremely wide models) omitted, including:
    * numerically stable softmax
    * softmax addition only being over rows
    * dot products being only n-1 additions (fused multiply-add exists anyway)
    """
    num_qheads = width / HEAD_SIZE
    num_kvheads = (
        2 * num_qheads / queries_per_group
    )

    embeddings = 0  # 0 if sparse lookup, backward FLOPs negligible

    attention = 2.0 * seq_len * (num_qheads + num_kvheads) * width * HEAD_SIZE
    attention += (
        3.5 * seq_len * (num_qheads + num_kvheads / 2) * HEAD_SIZE
    )  # RoPE, as implemented here/GPT-NeoX
    # score FLOPs are halved because causal => triangular mask => usable sparsity
    kq_logits = 1.0 * seq_len * seq_len * HEAD_SIZE * num_qheads
    softmax = 3.0 * seq_len * seq_len * num_qheads
    softmax_q_red = 2.0 * seq_len * seq_len * HEAD_SIZE * num_qheads
    final_linear = 2.0 * seq_len * width * HEAD_SIZE * num_qheads
    attn_bwd = (
        2.0 * attention
        + 2.5 * (kq_logits + softmax + softmax_q_red)
        + 2.0 * final_linear
    ) * depth
    attention += kq_logits + softmax + softmax_q_red + final_linear

    ffw_size = EXPAND_FACTOR * width
    dense_block = (
        6.0 * seq_len * width * ffw_size
    )  # three matmuls instead of usual two because of GEGLU
    dense_block += (
        10 * seq_len * ffw_size
    )  # 7 for other ops: 3 for cubic, two additions, two scalar mults
    dense_block += 2.0 * width * seq_len  # both/sandwich residual additions
    rmsnorm = 2 * 7.0 * width * seq_len

    final_rms_norm = 7.0 * width * seq_len  # one last RMSNorm
    final_logits = 2.0 * seq_len * width * vocab_size
    nonattn_bwd = 2.0 * (
        embeddings + depth * (dense_block + rmsnorm) + final_rms_norm + final_logits
    )
    forward_pass = (
        embeddings
        + depth * (attention + dense_block + rmsnorm)
        + final_rms_norm
        + final_logits
    )
    backward_pass = attn_bwd + nonattn_bwd  # flash attention

    return (forward_pass + backward_pass) / seq_len
```

