# OpenReview forum: "Gemstones: A Model Suite for Multi-Faceted Scaling Laws"
_NeurIPS.cc/2025/Conference — NeurIPS 2025 poster_

### Official Review · Reviewer_Kwid · 2025-06-03

**Clarity:** 3
**Significance:** 3
**Originality:** 2
**Rating:** 4
**Confidence:** 4

**Summary:**

The paper presents an empirical study on scaling laws. The setup is using the Dolma 1.7 dataset and training standard decoder-only LLMs with variable depth/width ratios. The contributions include:

1. Replicating Chinchilla's finding of keeping tokens/params constant when scaling (fig 1)
2. A new proposed fitting method using convex hulls.
3. A demonstration on the variability of fitting laws (table 1)
4. A demonstration that depth is better than width with constant flops (Figure 7,8)

**Questions:**

1. Are figures 4,5,6 novel results? It's not clear what is novel about them to me.
2. Could you try your convex hull method on other datasets? Does it consistently demonstrate better stability? If so, that's a good contribution that should be highlighted more.
3. Could you add a section with "recommendations for practitioners"? Currently there are lots of experiments, but the take-away message is less clear.
4. “then train at a constant learning rate, which we adjust for model size as described 137 in Appendix A.1”. Could you comment on why you are training with constant LR? Typically LLms are trained with cosine delay.

**Ethical Concerns:**

["NO or VERY MINOR ethics concerns only"]

**Final Justification:**

Nothing material has changed from my initial review that compels me to change the score.

**Quality:**

3

**Strengths And Weaknesses:**

Pros:
- The results on depth vs width are interesting and impactful.
- The demonstration of variability in fitting laws is useful.
- The proposed convex hull method could be impactful.
- The paper is well written.
- The authors give lots of details regarding experimental setup -- training data, model implementation etc.


Cons:
- Many of the results have limited novelty.
- The take-away message is not clear.




Minor nitpicks:
- Figure 1 doesn't convey any useful information. I'd recommend removing it. Everyone knows what width/depth is.
- Figure 3 has too bad aspect ratio, it needs more vertical space.
- Some references are missing, e.g. "Scaling Optimal LR Across Token Horizons".

---

> ### Author Rebuttal · Authors · 2025-07-30
>
> We thank the reviewer for their time and effort, and appreciate that they acknowledge the impact that our new convex hull approach and results on width and depth will have on the NeurIPS community.
>
>
> ## Figures 4,5 and 6
>
> To the best of our knowledge, Figure 4 is a novel result in a scientific setting. Figures 5 and 6 are demonstrating that our suite of models also extends to benchmark scaling laws. During the rebuttal period we have also conducted experiments fitting scaling laws over the datasets shown in Figure 4 to more effectively demonstrate the impact of using laws fitted on different datasets. Moreover, we extend our current benchmark scaling laws in Figures 4 and 5 by detailing their extrapolation when the 2b parameter models are held out. All of these experiments will be included in the camera ready version even though we cannot directly share them during the rebuttal process. In summary, we find that the formula proposed by Gadre et al. (fitted on all data in Figure 5) to be more robust to extrapolation than the formula proposed by Bhagia et al. (fitted on all data in Figure 6) with the Gadre et al. formula giving near perfect extrapolation to the 2 billion parameter models when fitted on all smaller models. With regards to fitting laws over multiple validation sets, we find that the fitted laws only vary slightly as the current Figure 4 implies.
>
>
> ## Convex Hull Method
>
> We agree the convex hull methodology has the potential to be highly impactful and widely used within the community. We hope that future research considers this method alongside the classical approach 1 fitting procedure and highlights how it can be used by practitioners to fit stable scaling laws with fewer data points.
>
>
> ## Recommendations
>
> We caution making sweeping recommendations for future practitioners due to the fragility we find to be inherent to scaling laws. Hence, our main recommendation is to be very careful of the pitfalls we highlight, and to make sure that the final configuration for training a larger “production” model mirrors the set of assumptions made when building the smaller models used to fit the scaling law. We also think our other contributions will also be useful to future practitioners: how width and depth impact training speed, how width and depth impact of benchmark accuracy versus validation loss, and how using the convex hull method for approach 1 can drastically reduce variability during fitting.
>
>
> ## Learning Rate Schedule
>
> Hägele at al. [1] suggest practitioners should use a warmup-stable-decay scheduler when fitting scaling laws as it allows intermediate checkpoints to be used for fitting and offers similar performance to a cosine decay scheduler. Appendix 1 details how we use a scalable initialisation and learning rate transfer to models of different shapes to be compared fairly in our study.
>
> In response to the reviewer’s valuable feedback, we have added extra references listed to our camera ready draft, and have made significant efforts to incorporate all other the feedback into the manuscript. If your primary concerns have been adequately addressed, and you do not have any other questions, please kindly consider raising your score as a stronger recommendation of acceptance.
>
> [1] Hägele A, Bakouch E, Kosson A, Von Werra L, Jaggi M. Scaling laws and compute-optimal training beyond fixed training durations. (Neurips 2024)

---

> > ### Comment · Reviewer_Kwid · 2025-08-02
> > **thanks**
> >
> > Thanks for your response, I will retain my score.

---

### Official Review · Reviewer_by1a · 2025-06-15

**Clarity:** 3
**Significance:** 2
**Originality:** 3
**Rating:** 3
**Confidence:** 4

**Summary:**

In this paper, the authors design new scaling law approach, including model selection, the choice of learning rate, and
107 curve fitting schemes. As a byproduct, the authors release the Gemstone, an open-source scaling law dataset, consisting of over 4000 checkpoints from transformers with up to 2 billion parameters and diverse architectural shapes.

**Questions:**

Please see Strengths And Weaknesses. I would expect to see answers for:

- if the model size exceeds 2B, will observations in this paper still hold? And at what size, the observations could deviate?

- If the model architecture changes to non-Gemma models, will observations in this paper still hold? If not, why?

- Please make the general discussion of the fragility and common pitfalls of prior scaling laws more detailed and clearly marked. It is also better to have some sort of side-by-side comparison with the proposed one. (for example, using a table/figure)

**Ethical Concerns:**

["NO or VERY MINOR ethics concerns only"]

**Final Justification:**

Thanks for your rebuttal and partially addressing my concerns. I have raised my score accordingly. However, the inherent problem of 2B model size and limitation of Gemma structure remains.

**Limitations:**

Yes.

**Quality:**

3

**Strengths And Weaknesses:**

(+) The authors open-sourced more than 4000 checkpoints cumulatively trained on over 10 trillion tokens.

(+) The paper is clearly written and easy to follow.

(-) The paper title is confusing. If Gemstone is a model suite, then it is more suitable for the DB track. Instead, it seems the authors propose a new scaling law approach. However, to this reviewer, it is unclear what is the takeaway from this new scaling law curve.

(-) The model size cut-off is 2 billion parameters, but is limited. Only the lightweight LLMs are of this size, and their standard version are usually much larger at example sizes of 7B, 8B, 13B or 70B. Such upper bound of 2 billion parameters also limit the study of scale up and "hitting the wall".

(-) Gemstone is loosely based on scaled-down variants of the Gemma architecture. It overlooks the massive LLM architectures used in other SOTA models such as LLAMA family, GPT family, and DeepSeek Family. This limits the generalizability of the observations.

(-) Scaling is usually discussed in the context of optimal performance of a given model size. It is expected that different architectures and hyperparameters may lead to a model of a given size that performs worse than the best known models at the same size. Therefore, this reviewer is not convinced by the motivation.

(-) The fragility and common pitfalls of prior scaling laws was not clearly spell out. It's better to make a more explicit comparision with the proposed method.

---

> ### Author Rebuttal · Authors · 2025-07-30
>
> We thank the reviewer for taking time to read our draft and their acknowledgement of the large contribution that our Gemstone model suite will be to the community.
>
>
> ## Our Choice of Track
>
> We believe our scientific contributions, including our analysis of the pitfalls of scaling laws and the scientific insights gained from varying the width and depth of the models, make this paper more appropriate for the main track. In addition, we introduce a convex hull fitting approach and analyze the impact of fitting scaling laws across different validation sets. Our results include several novel findings related to width and depth: increasing width decreases loss quicker than depth over time but slower over FLOPs, and increasing depth improves benchmark scores more than width under a fixed compute budget.
>
>
> ## Model Sizes Greater Than 2B
>
> We agree with the reviewer that studying larger models would be an interesting extension of our work, but as the compute required scales greatly with model size, we chose to train for longer token horizons rather than increasing model scale. We highlight that several scaling law papers have fitted on models with less than 2 billion parameters and produced results that generalise to much larger parameter counts [1,2,3]. Moreover, scaling laws fitted on small models have become commonplace in large SoTA model releases, predicting the behaviour of much larger models with extraordinary accuracy [4,5,6]. Hence, we believe our paper will be impactful for the NeurIPS community even thought it does not include models beyond 2b parameters.
>
>
> ## Non-Gemma Architectures
>
> In order to conduct a rigorous scientific study, we restrict some architectural considerations in order to analyze width and depth with high fidelity. Although many of our architectural choices are based upon Gemma models, we also take inspiration from Llama and Pythia models. Specifically, we take the convention of Llama models where the head dimension is always the embedding dimension divided by the number of heads and use the Pythia tokenizer in all experiments.
>
> We would also like to emphasize how our scaling law fragility results relate to these fixed architectural parameters. Since we find that many factors impact the fit of a scaling law we caution against transferring scaling laws without careful attention to the individual design decisions implicit in that scaling law. That said, we believe that in our setting our scaling laws would generalise as well as the ones shown in [2] and [3] who also fit on models of less than 2b parameters and generalise to much larger parameter counts with high accuracy. We also believe that our key methodological contributions such as the convex hull will transfer to other settings. Moreover, our findings such as the choice of aspect ratio to optimize loss with respect to time and to step count are widely impactful for all practitioners as we will open source all of our data. Finally, we hope this work drives extra consideration in other areas of architecture optimization such as expansion factor in transformer models and number of experts in a mixture of expert models as these are highly related to aspect ratio.
>
>
> ## Scaling
>
> We have to politely disagree with the reviewer and think scaling should be discussed in terms of optimal performance for a given compute budget (FLOPs), not model size. We believe the primary motivation of scaling laws is to optimize performance of larger models by training many smaller ones and fitting laws to the data those runs produce. We aim to offer the community a new angle to optimize performance at a given parameter count by leveraging our findings on how width and depth can be varied to achieve better performance. We note that it is common to overtrain models past any prescribed “compute optimal” point (Gadre et al.) and note how practitioners could also use our work to further understand how width and depth affects their FLOPs per training step and optimize over this dimension as well.
>
> We are aware that training models with varying widths and depths does lead to some models being suboptimal for a given FLOP budget and highlight how our convex hull approach is designed to stop this impacting our fitted laws. We also note the impactful additional information gained for our community by identifying which models are slightly suboptimal and our analysis of them throughout our paper. Finally, we highlight how the convex hull approach can also be used by the community in the future to fit high quality scaling laws with less data as shown in Figure 3.
>
>
> ## Common Pitfalls of Scaling Laws
>
> In the fragility and common pitfalls section, we are highlighting inherent fragility and pitfalls of all scaling laws. The asymptotic flatness of power law curves makes estimation of their parameters an ill-conditioned problem. Moreover, this is compounded when fitting (interpolating) what are already inherently noisy obervations from real training runs and we believe that our results show how this ill-conditioning can quickly impact predictions in many situations. We want to emphasize that all scaling laws in general should be considered in as close to their original setting and context as possible and practitioners should not expect them to transfer to significantly different settings without non-trivial effort. However, this is not a flaw of our study or methodology and to our knowledge, there is no specific proposed method to compare to that solves this variability problem entirely. With that said, in Figure 3, we do demonstrate how our convex hull method for the approach 1 fitting method (black crosses, blue line) compares to prior fitting approaches (red crosses, red line), and think that the reduction in the noise it yields is extremely promising.
>
> In response to your feedback, we have made significant efforts to improve our camera-ready version. If your primary concerns are at least somewhat addressed, and you have no further questions, please kindly consider raising your score.
>
> [1] Li M, Kudugunta S, Zettlemoyer L. (Mis) Fitting: A Survey of Scaling Laws. (ICLR 2025)
>
> [2] Kaplan J, McCandlish S, Henighan T, Brown TB, Chess B, Child R, Gray S, Radford A, Wu J, Amodei D. Scaling laws for neural language models. (arXiv)
>
> [3] Bhagia A, Liu J, Wettig A, Heineman D, Tafjord O, Jha AH, Soldaini L, Smith NA, Groeneveld D, Koh PW, Dodge J. Establishing task scaling laws via compute-efficient model ladders. (arXiv)
>
> [4] Grattafiori A, Dubey A, Jauhri A, Pandey A, Kadian A, Al-Dahle A, Letman A, Mathur A, Schelten A, Vaughan A, Yang A. The llama 3 herd of models. (arXiv)
>
> [5] Hu S, Tu Y, Han X, He C, Cui G, Long X, Zheng Z, Fang Y, Huang Y, Zhao W, Zhang X. Minicpm: Unveiling the potential of small language models with scalable training strategies. (arXiv)
>
> [6] Bi X, Chen D, Chen G, Chen S, Dai D, Deng C, Ding H, Dong K, Du Q, Fu Z, Gao H. Deepseek llm: Scaling open-source language models with longtermism. (arXiv)
>
> [7] Gadre SY, Smyrnis G, Shankar V, Gururangan S, Wortsman M, Shao R, Mercat J, Fang A, Li J, Keh S, Xin R. Language models scale reliably with over-training and on downstream tasks. (ICLR 2025)

---

> > ### Comment · Reviewer_by1a · 2025-08-05
> >
> > Thanks for your rebuttal and partially addressing my concerns. I have raised my score accordingly. However, the inherent problems of 2B model size and limitation of Gemma structure remains, which prevent me from raising further.

---

> ### Author Response · Authors · 2025-08-06
>
> We thank the reviewer for engaging with our rebuttal, taking the time to understand the nuances of our paper and increasing their score.
>
> Just because it is so relevant to both our work and this reviewer's comments, we wanted to bring an extremely recent release to the attention of the reviewer (posted during the rebuttal period). The trade off between width and depth has now also been analyzed by Zuo et al. [1] but in a different architecture family. They find strikingly similar trends in their analysis of hybrid state-space and transformer models in Sec 2.3.2 where they show that deeper models achieve lower loss at a fixed token budget but are less efficient to train. While our model architecture is itself a heterogeneous combination of Llama-style attention configuration, Gemma-style MLP layers, and the Pythia suite's vocabulary size, this new study closely corroborates our findings in a completely different architectural setting. Moreover, with what appear to be increased computational resources, they are able to demonstrate that a deeper 1.5b model can even match a 3b or 7b model of the same style, confirming that studies at the sub 2B scale may provide useful insight as to how to optimize performance at larger scales.
>
> We think that these results directly reinforce ours and that their models complement our own model suite. It is promising to see more thorough, scientifically reproducible investigations by our community, and this contemporary work has raised our confidence in the fact that the fully open source models, data, and rigorous analysis we present will serve as a valuable starting point for future work. We will be adding this citation and a short discussion to our camera ready draft.
>
> If this additional evidence and independent corroboration of our results has helped alleviate your remaining concerns, please consider raising your score as a stronger recommendation for acceptance.
>
> ---
>
> [1] Zuo J, Velikanov M, Chahed I, Belkada Y, Rhayem DE, Kunsch G, Hacid H, Yous H, Farhat B, Khadraoui I, Farooq M. Falcon-H1: A Family of Hybrid-Head Language Models Redefining Efficiency and Performance. (arXiv)

---

### Official Review · Reviewer_bWFc · 2025-07-01

**Clarity:** 2
**Significance:** 2
**Originality:** 1
**Rating:** 3
**Confidence:** 4

**Summary:**

This paper empirically studies the impact of various design choices on the outcome of scaling laws and highlights the sensitivity of scaling laws to these choices (in particular, the model depth, width, and the performance metrics being used). Alongside the empirical study, the paper also releases their checkpoints for more granular scaling analyses involving architectural and training hyperparameter variations.

**Questions:**

I have detailed most major questions in the weakness part in the previous section, and here are some remaining ones:
* In Table 2, which examines the variability of scaling laws without embedding parameters, Approach 3 appears to be quite reliable across different configurations. This raises the question of whether the observed "scaling law variability" is actually an artifact of the fitting variance, rather than a result of underlying design choices. Is that correct?
* How does the convex hull approach perform in Table 1 experiments?

**Ethical Concerns:**

["NO or VERY MINOR ethics concerns only"]

**Final Justification:**

While the author responses addresses some of my questions and I have raised my score to a 3, I still have several major concerns regarding the conceptual contributions and the rigor of the paper, and I think the paper needs major revision before it can be accepted:

**Limitations:**

Yes.

**Quality:**

2

**Strengths And Weaknesses:**

Strengths
* The paper studies the impact of some nuanced design choices on scaling laws that are sometimes understudied or overlooked in prior scaling laws, and highlights the sensitivity of scaling law outcomes to these choices. This study by itself could be interesting for practitioners, encouraging more careful studies of these nuanced choices.
* The paper releases their checkpoints for future scaling law analyses, which could be useful for practitioners with limited compute.


Weaknesses
* The conceptual contributions of this work seems limited:
  * The main conceptual takeaway of the empirical study – many decision choices could affect the outcome of scaling laws – have been extensively studied in previous works, and practitioners of scaling laws have been studying many of such nuances. It is not clear to me how this paper may change the way practitioners conduct scaling law analyses. I would encourage the authors to emphasize the key, discint contributions in the introduction.
  * The specific study of the impact of model aspect ratio on loss, downstream metrics, and training time could be useful, but seems constrained in scope and has been somewhat studied in prior works (as mentioned in the paper).
* The presentation of the paper could be improved:
  * In particular, the experiment section is composed of a stack of loosely connected experiments, which makes it a bit hard to understand the goal of each experiment and interpret their overall significance. I would encourage the author to restructure this section so that the motivation behind each experiment is clearly articulated in advance.
  * For the method section, the convex hull approach is only very briefly described without any formulations or details.
  * Some of the visualizations showing different model shapes (e.g., Figs. 4 and 8) are overloaded and difficult to interpret. I recommend grouping the annotations by marker and line style for each line group (e.g., by model depth and width) to improve clarity.
* Some empirical design or conclusion may not be rigorous:
  * There are not many unique training confirmations (e.g., 22 model shape variations) being used in the scaling law analyses, which make some analyses questionable:
    * In Figure 3, the comparison between different approaches may be confounded by the potential overfitting effect with limited data points. It would be much more convincing to include more data points and split them into train/test sets for robustness check.
    * In Table 1, the conclusion that the scaling law exponent is sensitive to various design choices is also confounded by the variability due to limited data points. (Pairwise) statistical testing needs to be included here to demonstrate significance.
    * The statement that “open-source more than 4000 checkpoints cumulatively trained on over 10 trillion tokens.” feels a overclaim since there are only a small set of unique training runs with 350B training tokens.
  * In the cooldown phase, the model is trained on an additional 10% of the total tokens it has already seen during training. This data repetition seems unusual and may affect the analyses. Could you clarify the motivation behind this choice?

---

> ### Author Rebuttal · Authors · 2025-07-30
>
> We thank the reviewer for their thoughtful review of our work and appreciate that they have noted the impact that open sourcing the Gemstone model suite will have on the scientific community we are a part of. We also hope that our release will enable more researchers to participate in nuanced discussions on width and depth, which the reviewer notes have previously been understudied.
>
>
> ## Conceptual Contributions
>
> We acknowledge we are not the first to see variability in scaling law analyses, and discuss this in our related work section. However, we do believe that showcasing this variability along both previously studied axes as well as novel ones in a single controlled setting is of value to the community as scaling laws are widely used for planning model training runs. In particular, we think that a large-scale study of width and depth provided by this work along paired with scaling law analyses represents a non-trivial addition to the open source literature. For example, in the Gemma 2 report [1], Table 9 presents a single result on how depth increases benchmark performance more than width, but few details about the experiment are included. In contrast, our work corroborates this in a scientific setting and open source all results for the community to build upon. We also emphasize our other novel, impactful contributions that the reviewer already notes, such as finding width and depth to affect loss differently over time vs. FLOPs, analyzing transfer of scaling laws over different validation datasets, and introducing the convex hull methodology.
>
>
> ## The Convex Hull Approach
>
> We provide a mathematical definition below, loosely based on the wikipedia entry for reconstructing functions from epigraphs, and have added this to our camera ready draft:
>
> We can define the set of points we have to fit on as FLOPs/GPU hours $(x)$, loss value $(L)$ pairs as $\mathcal{D} = \\{ (x_i, L_i) \\}_{i=1}^n \subset \mathbb{R}^2$
>
> Let $ \text{conv}(\mathcal{D}) $ denote the convex hull, the linear interpolation of any two points in $\mathcal{D}$:
> $$
> \text{conv}(\mathcal{D}) = \left\\{ \sum_{i=1}^{n} \lambda_i (x_i, L_i) \ \middle|\ \lambda_i \geq 0,\ \sum_{i=1}^{n} \lambda_i = 1 \right\\}
> $$
>
> Define $\hat{L}(x) = \min(\\{y | (x,y) \in \mathcal{D}\\})$ as the lower convex hull is the graph of this new function:
> $ \\{(x, \hat{L}(x)) | x \in \text{Dom}(\hat{L})\\} $
>
> We think the easiest visualisation of this is in Figure 7 where the red line is the convex hull, the colored crosses are the vertices and the grey lines are all possible points in the dataset. The exact code to generate the convex hull is in our supplementary material, in the function `plotters.figure_2_correct_data.get_resource_hull`
>
> ## Rigour
>
> The reviewer has noticed a key merit of our paper in that we present two distinct sets of contributions. One part of our work looks at the established set of design choices one must make when fitting scaling laws and the other considers varying the shapes of the models we train along dimensions of width and depth.
>
> We remark that many impactful scaling law studies train fewer distinct model shapes than we do, such as Porian et al. [4] who train 16 distinct model shapes and even one of the largest studies, Chinchilla [2], only considers 50 distinct model shapes. To fit a scaling law, we need data points which consist of a compute budget (often made up of a model parameter count and number of training tokens) and loss values. Some prior work such as Chinchilla [2], Kaplan [3] and Porian [4] choose to do this by running many independent training runs as they use a cosine learning rate scheduler. To reduce cost and increase efficiency, we follow Hägele et al. [5] who propose practitioners should use a warmup-stable-decay learning rate scheduler so that all intermediate checkpoints in the stable regime can be used during fitting, this means we can utilise all 4000 of our checkpoints in different ways when fitting scaling laws much more effectively and train for much larger token counts. Our 4000 checkpoints can be split into 3 categories {main, cooldown, lr ablation}; each can use all points to fit a scaling law. We also highlight that in Table 1 we see fragility over each of these subsets, inferring this is a repeatable pattern. We also refer to our related works section to highlight some fragility in scaling laws has been seen in prior work as it is inherent to all scaling laws due to the asymptotic flatness of power law curves; we corroborate and extend these results leveraging controlled experimental training of the Gemstone suite.
>
>
> ## Cooldown
>
> 10% here is referring to token counts—_there is no data repetition during training run in any of our experiments_. For example, a model trained for 40b tokens and then cooled down, would be cooled down on 4b _new_ tokens. However, it is worth noting that the 4b additional tokens the model would see during cooldown are actually the next 4b tokens that the model would have trained on had the learning rate remained constant. This design allows for a controlled comparison where two “hot” and “cooled down” checkpoints can be compared such the _only_ difference is whether or not the most recent 10% of tokens were trained on at a constant lr, or while the lr was linearly decaying.
>
>
> ## Reducing Variance During Fitting
>
> We would like to highlight the reduced variance and better fit found by our convex hull approach as demonstrated in Figure 3 and utilized throughout the paper including the experiments behind Tables 1 and 2. In addition, as shown in Figure 19, we also check that our grid search size and delta in the Huber loss are in a low variance regime when fitting Approach 3 laws to limit the impact of fitting noise on our results. During the rebuttal period, we have also conducted a leave-one-out analysis over the 22 models and added this to the camera ready draft appendix so that the reader can visually contextualize fitting variability caused by model selection.
>
> As we have made significant efforts to incorporate all feedback into our camera-ready version, including addressing concerns over readability of figures, if you feel that your questions have been sufficiently addressed, please kindly consider raising your score.
>
> [1] Team G. Gemma 2: Improving open language models at a practical size. (arXiv)
>
> [2] Hoffmann J, Borgeaud S, Mensch A, Buchatskaya E, Cai T, Rutherford E, Casas DD, Hendricks LA, Welbl J, Clark A, Hennigan T. Training compute-optimal large language models. (arXiv)
>
> [3] Kaplan J, McCandlish S, Henighan T, Brown TB, Chess B, Child R, Gray S, Radford A, Wu J, Amodei D. Scaling laws for neural language models. (arXiv)
>
> [4] Porian T, Wortsman M, Jitsev J, Schmidt L, Carmon Y. Resolving discrepancies in compute-optimal scaling of language models. (NeurIPS 2024)
>
> [5] Hägele A, Bakouch E, Kosson A, Von Werra L, Jaggi M. Scaling laws and compute-optimal training beyond fixed training durations. (Neurips 2024)

---

> > ### Comment · Reviewer_bWFc · 2025-08-03
> >
> > Thank you for your detailed response, and I appreciate the effort you've put into it. While it addresses some of my questions and I will raise my score to a 3, I still have several major concerns about the paper and think the paper needs major revision before it can be accepted:
> >
> > * **Conceptual contributions**: It remains unclear to me what the core insights of the paper are and how they might change the way practitioners study scaling laws. While the response lists a few findings, they mostly seem to be investigations of empirical design choices with limited conceptual significance. For readers already familiar with scaling law analyses, many of these observations—such as the brittleness of scaling laws—are well-known from prior work. Moreover, practitioners typically explore design choices that are relevant to their specific context, so it’s unclear what new guidance this paper offers.
> >
> > * **Rigor**, particularly concerning variance and overfitting due to limited model shapes:
> >
> >   * I don't find the argument that “previous works also don’t use many data points with distinct shapes” particularly compelling, as those works have different goals. This paper positions the study of model shape as a key contribution, so the supporting empirical evidence should be stronger and more carefully validated.
> >   * My concern was the variance introduced by the limited number of data points used to fit scaling laws, not just hyperparameter sensitivity mentioned in the response. I still believe it is critical to isolate these effects through holdout validation and statistical testing, as suggested in my original review.

---

> > > ### Author Response · Authors · 2025-08-05
> > >
> > > We would like to thank the reviewer for engaging with our rebuttal and acknowledging our efforts.
> > >
> > > We acknowledge the challenge of variance when fitting in scaling laws is inherent and possibly exacerbated when also varying other hyperparameters, such as width and depth, we hope that the convex hull method is a good first step in addressing this within the community.

---

### Official Review · Reviewer_2Aaa · 2025-07-02

**Clarity:** 3
**Significance:** 3
**Originality:** 3
**Rating:** 5
**Confidence:** 4

**Summary:**

This paper is an empirical study investigating the fragility of neural scaling laws. The authors argue that existing prescriptions are often unreliable as they are highly sensitive to the experimental design process.

To enable a more robust analysis, the authors introduce "Gemstones," an open-source suite of over 4,000 Transformer checkpoints (up to 2B parameters) with diverse width/depth ratios and training configurations. Using this dataset, they demonstrate that:

- Scaling laws are highly sensitive to the specific models and hyperparameters used for fitting. In particular, I really like this point "There are many decisions to make when choosing points to fit scaling laws that significantly change the slope of the law"

- A critical width-vs-depth trade-off exists: deeper models are more FLOP-efficient, while wider models are more time-efficient in terms of wall-clock hours (note that this is workflow dependent)

The main contribution is the model suite itself, which provides a valuable resource for the community to study scaling phenomena without prohibitive compute costs.

I thank the authors for doing a great service to the open research community. To further increase the paper's impact, I would encourage them to also release a Colab notebook to aggregate and visualize the experimental data (e.g., FLOPs vs. loss, downstream performance), which would make it easier for users to access and build upon their findings.

**Questions:**

see above.

**Ethical Concerns:**

["NO or VERY MINOR ethics concerns only"]

**Limitations:**

yes

**Paper Formatting Concerns:**

va

**Quality:**

3

**Strengths And Weaknesses:**

Strengths

- Clarity and High-Quality Presentation: The paper is well-written and easy to follow. The authors should be commended for explaining their design choices in detail.

- Insightful Ablation Studies: The width/depth ratio analysis is a particularly interesting and useful contribution. The authors could strengthen this by comparing their findings against Kaplan's original scaling law paper and referencing the discussions in Allen-Zhu's "Physics of LLMs."

- Effective Demonstration of Fragility: Table 1 provides a very clear and compelling demonstration of the variability in fitting scaling laws, successfully highlighting the fragility of current methods.

Weaknesses

- Limited Model Scale: The primary weakness is that the study is limited to models up to 2B parameters. While an analysis at a larger scale (e.g., 10B) would be more impactful, it is completely understandable that this is constrained by the significant computational cost.

---

> ### Author Rebuttal · Authors · 2025-07-30
>
> We thank the reviewer for their time and insightful comments on our draft. We appreciate their acknowledgement of the fact that the Gemstone model suite itself represents a significant contribution, and are also encouraged by the fact that they find our detailed analysis of the fragility of scaling laws and our findings on the trade off of width and depth when training transformer language models interesting.
>
>
> ## Larger Scale Models
>
> We too would like to scale to larger models but unfortunately the compute required increases greatly with model size and so training models larger than 2 billion parameters was simply not possible within our compute budget. We still feel that our work will have a significant impact on the scaling laws community as others have shown laws fitted on models with up to 2b parameters can be highly informative scaling laws [1,2,3]. Moreover, in the Gemma 2 report [3], Table 9 already corroborates our prediction that depth improves benchmark performance more than width; our work demonstrates this trend in an open, scientific and reproducible manner for the NeurIPS community. Hence, we believe our work can still be very impactful for the NeurIPS community despite the fact that the models we release are smaller than those that industry labs often train.
>
>
> ## Colab Notebook
>
> We agree that in addition to the model suite and fitting datasets, an interactive colab notebook could greatly increase the accessibility of our data for the community and thereby elevate the impact of the work. We are happy to invest effort into this while preparing the camera ready manuscript for publication.
>
> ## More Nuanced Discussion Points
>
> We appreciate the reviewer pointing out a potential connection between our work and Allen-Zhu's "Physics of LLMs." We have updated our camera ready draft to include a discussion these nuanced links focusing on how design decisions in “clean room” scientific settings can transfer to larger scale and more realistic experiments.
>
> As we have made significant efforts to incorporate all feedback into our camera-ready version, if you do not have any other questions we can address, please kindly consider raising your score as an even stronger recommendation of acceptance.
>
> [1] Li M, Kudugunta S, Zettlemoyer L. (Mis) Fitting: A Survey of Scaling Laws. (ICLR 2025)
>
> [2] Kaplan J, McCandlish S, Henighan T, Brown TB, Chess B, Child R, Gray S, Radford A, Wu J, Amodei D. Scaling laws for neural language models. (arXiv)
>
> [3] Bhagia A, Liu J, Wettig A, Heineman D, Tafjord O, Jha AH, Soldaini L, Smith NA, Groeneveld D, Koh PW, Dodge J. Establishing task scaling laws via compute-efficient model ladders. (arXiv)
>
> [4] Team G. Gemma 2: Improving open language models at a practical size. (arXiv)

---

> > ### Comment · Reviewer_2Aaa · 2025-08-01
> > **re**
> >
> > Thank you for the reply. This is a good paper, and I will maintain my score in support of its acceptance. I would, however, appreciate it if the authors could make the experimental data more accessible, for instance, through a Colab notebook.

---

### Official Review · Reviewer_shdg · 2025-07-05

**Clarity:** 3
**Significance:** 3
**Originality:** 3
**Rating:** 5
**Confidence:** 2

**Summary:**

This paper studies scaling laws for pre-training while going beyond narrow hyper-parameter choices and varying both depth and width. They futhur adapt the typical formulation of scaling laws to be more robust to biases and sparse data regions in the fitting data. The authors Intermidairy checkpoints are also released at frequent intervals to aid in further open-science study of these laws.

**Questions:**

- Since intermidiary checkpoints are also used in the fitting, does this bias the laws? how sensitive are the laws with respect to the frequency of the included checkpoints?
- If you were able given resource and time-constraints, etc. to ablate another axis, which of the ones listed in limitations would you priotize and why? I think it'd would be an appreciated piece of information if included in the limitations section with more detail.

**Ethical Concerns:**

["NO or VERY MINOR ethics concerns only"]

**Limitations:**

yes

**Quality:**

3

**Strengths And Weaknesses:**

Strengths:
- The paper is genrally well-written
- The experimental depth of the work requires singificant compute and resources which is not widely accessible to most researchers. The open-sourcing of the many model checkpoints will be greatly valuable to the community.
- The paper covers both different optimization, and architectural choices in deriving the laws.

Weaknesses:
- The work would have benefited from having a larger scale of models, exclusively reserved for testing laws derived at a smaller scale. My understanding is that all datapoints up to 2B scale are used in fitting the loss, correct? It would be good to test the laws on a held out model size using the optimal ratios dervied from the fittings.

---

> ### Author Rebuttal · Authors · 2025-07-30
>
> We thank the reviewer for their thoughtful comments and their acknowledgement of the multi-faceted contributions that the Gemstone model suite will make to the scientific community.
>
> ## Larger Scale Models
>
> We too would like to scale to larger models but unfortunately the compute required increases greatly with model size. Training models larger than 2 billion parameters was simply not possible within our compute budget. Kaplan et al. [1], one of the most widely known scaling laws, also only scale to 1.5 billion non-embedding parameter models, so we feel our work can still have a significant impact on the scaling laws community at this scale. Moreover, Bhagia et al. [2] only use models of up to 2b parameters to predict the performance of 6b and 14b parameter models. Finally, in the Gemma 2 report [3], Table 9, already corroborates our prediction that depth improves benchmark performance more than width; our work demonstrates this trend in an open, scientific and reproducible manner for the NeurIPS community. Hence, we believe our paper will be very impactful to the community.
>
> During the rebuttal period, we have updated our manuscript to include benchmark scaling laws where we fit to models less than 2b parameters and check the predictions against the actual data for our 2b parameter models observing high agreement. However, the rebuttal process constraints mean that we cannot share the new plots directly, so we will simply summarize the results here and then include them in the camera ready copy. We find that the formula proposed by Gadre et al. (fitted on all data in Figure 5) to be more robust to extrapolation than the formula proposed by Bhagia et al. (fitted on all data in Figure 6). The Gadre et al. formula yields near perfect extrapolation to the 2 billion parameter models when fitted on all models with less than 2b parameters.
>
> ## Using Intermediate Checkpoints
>
> Hägele at al. [4] suggest practitioners should use a warmup-stable-decay scheduler when fitting scaling laws as it allows intermediate checkpoints to be used for fitting scaling laws and offers similar performance to a cosine decay scheduler. The laws we present are fit based on checkpoints taken every 10b training tokens, but we also fit laws on checkpoints every 2b tokens and found these to be the same; hence for efficiency we fit on checkpoints taken every 10b tokens. We have included this ablation fitting on checkpoints every 2b tokens in our camera ready appendix, showing little to no difference to the laws fitted on checkpoints every 10b tokens.
>
> ## What would we do next?
>
> Given additional time and computational resources, we would first look at the impact of the expansion factor of the transformer, as this is most closely related to the aspect ratio and it would be interesting to compare and contrast to our current findings.
>
> We thank the reviewer again for their effort in engaging with our work so far. We believe that your feedback has been valuable in preparing a more camera-ready version of our manuscript. If you find our responses satisfactory and there are no further questions we can address, please kindly consider moving your score in the direction of an even stronger recommendation of acceptance.
>
> [1] Kaplan J, McCandlish S, Henighan T, Brown TB, Chess B, Child R, Gray S, Radford A, Wu J, Amodei D. Scaling laws for neural language models. (arXiv)
>
> [2] Bhagia A, Liu J, Wettig A, Heineman D, Tafjord O, Jha AH, Soldaini L, Smith NA, Groeneveld D, Koh PW, Dodge J. Establishing task scaling laws via compute-efficient model ladders. (arXiv)
>
> [3] Team G. Gemma 2: Improving open language models at a practical size. (arXiv)
>
> [4] Hägele A, Bakouch E, Kosson A, Von Werra L, Jaggi M. Scaling laws and compute-optimal training beyond fixed training durations. (Neurips 2024)

---

### Note · Authors · 2025-08-12

We would like to reiterate our thanks to all reviewers for their thoughtful engagement with us during the rebuttal period. Below we simply highlight some of the most important points from these fruitful discussions:

## New Experiments

(charts to appear in the camera ready copy)

1. In response to **Kwid** and **shdg** we extended our benchmark scaling laws in Fig. 4 and 5 by testing their extrapolation when the 2b parameter models are held out. We find that the formula proposed by Gadre et al. (fit on data in Fig. 5) is more robust to extrapolation than the formula proposed by Bhagia et al. (fit on data in Fig. 6) with the former providing near perfect extrapolation to the 2b parameter models.

2. In response to **bWFc** we have run validations where we leave out each model shape and fit over the rest of them. It is visually apparent in these charts that there is little meaningful variance in the fitted scaling law when one model is left out.

3. In response to **Kwid** we fit scaling laws over the multiple validation sets presented in Fig. 4 and find that fits vary only slightly. We hope that practitioners can rely on this robustness if their validation sets differ from the ones used to fit prior laws.



## Clarifications

1. **Variance in Fitting:** Table 1 reports the effect that various interventions have on scaling law prescriptions quantifying their fragility in a way that is easy to understand. We believe this demonstrates a fundamental challenge about the scaling laws paradigm itself: the asymptotic flatness of power law curves makes trying to fit a scaling law an ill-conditioned optimization problem. Our novel convex hull approach is specifically crafted to reduce this variance and our results suggest that when optimal points are sparse, our approach can be used to obtain a more reliable fit (red vs black crosses in Fig. 3).

2. **Generalization of Analysis to Other Models:** While their tech report provides few details, our results corroborate the trend reported by the Gemma 2 team (9b scale) that deeper models can outperform shallower ones. Furthermore, [1], released during the rebuttal period, finds similar trends at the 1.5b scale for hybrid transformer + state space models. We believe that this is all evidence that our model suite and analysis will be impactful even if future work differs slightly in terms of experimental settings, architecture, or parameter counts.

[1] "Falcon-H1: A Family of Hybrid-Head Language Models" arXiv:2507.22448

---

### Decision · Program_Chairs · 2025-09-17

**Decision:**

Accept (poster)

**Comment:**

This is a good empirical scaling study, which covers Gemma-LM scaling.

The main contributions (and positives) are the following:

1) Depth vs Width alternatives. Most previous studies stay within a very tight range of depth/width pattern (Figure 2). This paper shows 2 interesting features on a depth-width tradeoff. Moving off this line tend to produce flop-optimal models. However wider models are more wallclock-time optimal.
2) They provide a large family of opensourced checkpoints for exploration. Extensive details on setup: initialization, LR choices, etc. are appreciated for full reproducibility (which is famously lacking in some scaling law papers).

They have acknowledged limitations (model size <2B), restricted to Gemma-family models.

I believe on balance, this is a good paper; I believe the referees are generally positive about the content and presentation, with only minor suggestions within the scope of paper (for example, scaling this work is obviously desirable but introduces resource demands which may not feasible; further, at current scales, it demonstrates a trend already).

(Accept)